# A Unified Approach for Maximizing Continuous DR-submodular Functions

**Mohammad Pedramfar**
Purdue University
mpedramf@purdue.edu

**Christopher John Quinn**
Iowa State University
cjquinn@iastate.edu

**Vaneet Aggarwal**
Purdue University
vaneet@purdue.edu

## Abstract

This paper presents a unified approach for maximizing continuous DR-submodular functions that encompasses a range of settings and oracle access types. Our approach includes a Frank-Wolfe type offline algorithm for both monotone and non-monotone functions, with different restrictions on the general convex set. We consider settings where the oracle provides access to either the gradient of the function or only the function value, and where the oracle access is either deterministic or stochastic. We determine the number of required oracle accesses in all cases. Our approach gives new/improved results for nine out of the sixteen considered cases, avoids computationally expensive projections in three cases, with the proposed framework matching performance of state-of-the-art approaches in the remaining four cases. Notably, our approach for the stochastic function value-based oracle enables the first regret bounds with bandit feedback for stochastic DR-submodular functions.

## 1 Introduction

The problem of optimizing DR-submodular functions over a convex set has attracted considerable interest in both the machine learning and theoretical computer science communities [2, 5, 16, 26]. This is due to its many practical applications in modeling real-world problems, such as influence/revenue maximization, facility location, and non-convex/non-concave quadratic programming [3, 9, 18, 15, 20]. as well as more recently identified applications like serving heterogeneous learners under networking constraints [20] and joint optimization of routing and caching in networks [21].

Numerous studies investigated developing approximation algorithms for constrained DR-submodular maximization, utilizing a variety of algorithms and proof analysis techniques. These studies have addressed both monotone and non-monotone functions and considered various types of constraints on the feasible region. The studies have also considered different types of oracles—gradient oracles and value oracles, where the oracles could be exact (deterministic) or stochastic. Lastly, for some of the aforementioned offline problem settings, some studies have also considered analogous online optimization problem settings as well, where performance is measured in regret over a horizon. This paper aims to unify the disparate offline problems under a single framework by providing a comprehensive algorithm and analysis approach that covers a broad range of setups. By providing a unified framework, this paper presents novel results for several cases where previous research was either limited or non-existent, both for offline optimization problems and extensions to related stochastic online optimization problems.

This paper presents a Frank-Wolfe based meta-algorithm for (offline) constrained DR-submodular maximization where we could only query within the constraint set, with sixteen variants for sixteen problem settings. The algorithm is designed to handle settings where (i) the function is monotone

---

This work was supported in part by the National Science Foundation under grants CCF-2149588 and CCF-2149617, and Cisco, Inc.

37th Conference on Neural Information Processing Systems (NeurIPS 2023).

Table 1: Offline DR-submodular optimization results.

| $F$ | Set | Oracle | Setting | Reference | Appx. | Complexity |
|---|---|---|---|---|---|---|
| Monotone | $0 \in \mathcal{K}$ | $\nabla F$ | det. | [4], (*) | $1 - 1/e$ | $O(1/\epsilon)$ |
| | | | stoch. | [23], (*) | $1 - 1/e$ | $O(1/\epsilon^3)$ |
| | | | | [32] ‡ | $1 - 1/e$ | $O(1/\epsilon^2)$ |
| | | $F$ | det. | This paper | $1 - 1/e$ | $O(1/\epsilon^3)$ |
| | | | stoch. | This paper | $1 - 1/e$ | $O(1/\epsilon^5)$ |
| | general† | $\nabla F$ | det. | [16] ‡ | $1/2$ | $O(1/\epsilon)$ |
| | | | | This paper | $1/2$ | $\tilde{O}(1/\epsilon)$ |
| | | | stoch. | [16]‡ | $1/2$ | $O(1/\epsilon^2)$ |
| | | | | This paper | $1/2$ | $\tilde{O}(1/\epsilon^3)$ |
| | | $F$ | det. | This paper | $1/2$ | $\tilde{O}(1/\epsilon^3)$ |
| | | | stoch. | This paper | $1/2$ | $\tilde{O}(1/\epsilon^5)$ |
| Non-Monotone | d.c. | $\nabla F$ | det. | [3], (*) | $1/e$ | $O(1/\epsilon)$ |
| | | | stoch. | [23], (*) | $1/e$ | $O(1/\epsilon^3)$ |
| | | $F$ | det. | This paper | $1/e$ | $O(1/\epsilon^3)$ |
| | | | stoch. | This paper | $1/e$ | $O(1/\epsilon^5)$ |
| | general | $\nabla F$ | det. | [12] | $\frac{1-h}{3\sqrt{3}}$ | $O(e^{\sqrt{dL/\epsilon}})$ |
| | | | | [11] | $\frac{1-h}{4}$ | $O(e^{\sqrt{dL/\epsilon}})$ |
| | | | | [10], (*) | $\frac{1-h}{4}$ | $O(1/\epsilon)$ |
| | | | stoch. | This paper | $\frac{1-h}{4}$ | $O(1/\epsilon^3)$ |
| | | $F$ | det. | This paper | $\frac{1-h}{4}$ | $O(1/\epsilon^3)$ |
| | | | stoch. | This paper | $\frac{1-h}{4}$ | $O(1/\epsilon^5)$ |

This table compares the different results for the number of oracle calls (complexity) *within the feasible set* for DR-submodular maximization. Shaded rows indicate problem settings for which our work has the **first guarantees** or **beats the SOTA**. The rows marked with a blue star (*) correspond to cases where Algorithm 2 **generalizes the corresponding algorithm** and therefore has the same performance. The different columns enumerate properties of the function, the convex feasible region (downward-closed, includes the origin, or general), and the oracle, as well as the approximation ratios and oracle complexity (the number of queries needed to achieve the stated approximation ratio with at most $\epsilon > 0$ additive error). We have $h := \min_{\mathbf{x} \in \mathcal{K}} \|x\|_\infty$. (See Appendix B regarding [23] and [32]). † when the oracle can be queried for any points in $[0,1]^d$ (even outside the feasible region $\mathcal{K}$), the problem of optimizing monotone DR-submodular functions over a general convex set simplifies — [4] and [23] achieve the same ratios and complexity bounds as listed above for $0 \in \mathcal{K}$; [8] can achieve an approximation ratio of $1 - 1/e$ with the $O(1/\epsilon^3)$ and $O(1/\epsilon^5)$ complexity for exact and stochastic value oracles respectively. ‡ [16] and [32] use gradient ascent, requiring potentially computationally expensive projections.

or non-monotone, (ii) the feasible region is a downward-closed (d.c.) set (extended to include $0$ for monotone functions) or a general convex set, (iii) gradient or value oracle access is available, and (iv) the oracle is exact or stochastic. Table 1 enumerates the cases and corresponding results on oracle complexity (further details are provided in Appendix A). We derive the first oracle complexity guarantees for nine cases, derive the oracle complexity in three cases where previous result had a computationally expensive projection step [32, 16] (and we obtain matching complexity in one of these), and obtain matching guarantees in the remaining four cases. In addition to proving approximation ratios and oracle complexities for several (challenging) settings that are the first or improvements over the state of the art, the *technical novelties of our approach* include:

(i) A new construction procedure of a shrunk constraint set that allows us to work with lower dimensional feasible sets when given a value oracle, resulting in the first results on general lower dimensional feasible sets given a value oracle.

(ii) The first Frank-Wolfe type algorithm for analyzing monotone functions over a general convex set for any type of oracle, where only feasible points can be queried.

(iii) Shedding light on a previously unexplained gap in approximation guarantees for monotone DR-submodular maximization. Specifically, by considering the notion of query sets and assuming that the oracles can only be queries within the constraint set, we divide the class of

monotone submodular maximization into monotone submodular maximization over convex sets containing the origin and monotone submodular maximization over general convex sets. Moreover, we conjecture that the $1/2$ approximation coefficient, which has been considered sub-optimal in the literature, is optimal when oracle queries can only be made within the constraint set. (See Appendix B for more details.)

Furthermore, we also consider online stochastic DR-submodular optimization with bandit feedback, where an agent sequentially picks actions (from a convex feasible region), receives stochastic rewards (in expectation a DR-submodular function) but no additional information, and seeks to maximize the expected cumulative reward. Performance is measured against the best action in expectation (or a near-optimal baseline when the offline problem is NP-hard but can be approximated to within $\alpha$ in polynomial time), the difference denoted as expected $\alpha$-regret. For such problems, when only bandit feedback is available (it is typically a strong assumption that semi-bandit or full-information feedback is available), the agent must be able to learn from stochastic value oracle queries over the feasible actions action. By designing offline algorithms that only query feasible points, we made it possible to convert those offline algorithms into online algorithms. In fact, because of how we designed the offline algorithms, we are able to access them in a black-box fashion for online problems when only bandit feedback is available. Note that previous works on DR-submodular maximization with bandit feedback in monotone settings (e.g. [31], [25] and [30]) explicitly assume that the convex set contains the origin.

For each of the offline setups, we extend the offline algorithm (the respective variants for stochastic value oracle) and oracle query guarantees to provide algorithms and $\alpha$-regret bounds in the bandit feedback scenario. Table 2 enumerates the problem settings and expected regret bounds with bandit and semi-bandit feedback. The key contributions of this work can be summarized as follows:

**1.** This paper proposes a unified approach for maximizing continuous DR-submodular functions in a range of settings with different oracle access types, feasible region properties, and function properties. A Frank-Wolfe based algorithm is introduced, which compared to SOTA methods for each of the sixteen settings, achieves the best-known approximation coefficients for each case while providing (i) the first guarantees in nine cases, (ii) reduced computational complexity by avoiding projections in three cases, and (iii) matching guarantees in remaining four cases.

**2.** In particular, this paper gives the first results on offline DR-submodular maximization (for both monotone and non-monotone functions) over general convex sets and even for downward-closed convex sets, when only a value oracle is available over the feasible set. Most prior works on offline DR-submodular maximization require access to a gradient oracle.

**3.** The results, summarized in Table 2, are presented with two feedback models—bandit feedback where only the (stochastic) reward value is available and semi-bandit feedback where a single stochastic sample of the gradient at the location is provided. This paper presents the first regret analysis with bandit feedback for stochastic DR-submodular maximization for both monotone and non-monotone functions. For semi-bandit feedback case, we provide the first result in one case, improve the state of the art result in two cases, and gives the result without computationally intensive projections in one case.

Table 2: Online stochastic DR-submodular optimization.

| $F$ | Set | Feedback | Reference | Coef. $\alpha$ | $\alpha$-Regret |
|---|---|---|---|---|---|
| Monotone | $0 \in \mathcal{K}$ | $\nabla F$ | [6]†, | $1/e$ | $O(T^{2/3})$ |
| | | | This paper | $1-1/e$ | $O(T^{3/4})$ |
| | | $F$ | This paper | $1-1/e$ | $O(T^{5/6})$ |
| | general | $\nabla F$ | [16] ‡ | $1/2$ | $O(T^{1/2})$ |
| | | | This paper | $1/2$ | $\tilde{O}(T^{3/4})$ |
| | | $F$ | This paper | $1/2$ | $\tilde{O}(T^{5/6})$ |
| Non-mono. | d.c. | $\nabla F$ | This paper | $1/e$ | $O(T^{3/4})$ |
| | | $F$ | This paper | $1/e$ | $O(T^{5/6})$ |
| | general | $\nabla F$ | This paper | $\frac{1-h}{4}$ | $O(T^{3/4})$ |
| | | $F$ | This paper | $\frac{1-h}{4}$ | $O(T^{5/6})$ |

This table compares the different results for the expected $\alpha$-regret for online stochastic DR-submodular maximization for the under bandit and semi-bandit feedback. Shaded rows indicate problem settings for which our work has the **first guarantees** or **beats the SOTA**. We have $h := \min_{\mathbf{x}\in\mathcal{K}} \|x\|_\infty$. † the analysis in [6] has an error (see the supplementary material for details). ‡ [16] uses gradient ascent, requiring potentially computationally expensive projections.

**Related Work:** The key related works are summarized in Tables 1 and 2. We briefly discuss some key works here; see the supplementary materials for more discussion. For online DR-submodular optimization with bandit feedback, there has been some prior works in the adversarial setup [31, 33, 25, 30] which are not included in Table 2 as we consider the stochastic setup. [31] considered monotone DR-submodular functions over downward-closed convex sets and achieved $(1 - 1/e)$-regret of $O(T^{8/9})$ in adversarial setting. This was later improved by [25] to $O(T^{5/6})$. [30] further improved the regret bound to $O(T^{3/4})$. However, it should be noted that they use a convex optimization subroutine at each iteration which could be even more computationally expensive than projection. [33] considered non-monotone DR-submodular functions over downward-closed convex sets and achieved $1/e$-regret of $O(T^{8/9})$ in adversarial setting.

## 2 Background and Notation

We introduce some basic notions, concepts and assumptions which will be used throughout the paper. For any vector $\mathbf{x} \in \mathbb{R}^d$, $[\mathbf{x}]_i$ is the $i$-th entry of $\mathbf{x}$. We consider the partial order on $\mathbb{R}^d$ where $\mathbf{x} \leq \mathbf{y}$ if and only if $[\mathbf{x}]_i \leq [\mathbf{y}]_i$ for all $1 \leq i \leq d$. For two vectors $\mathbf{x}, \mathbf{y} \in \mathbb{R}^d$, the *join* of $\mathbf{x}$ and $\mathbf{y}$, denoted by $\mathbf{x} \vee \mathbf{y}$ and the *meet* of $\mathbf{x}$ and $\mathbf{y}$, denoted by $\mathbf{x} \wedge \mathbf{y}$, are defined by

$$\mathbf{x} \vee \mathbf{y} := (\max\{[\mathbf{x}]_i, [\mathbf{y}]_i\})_{i=1}^d \quad \text{and} \quad \mathbf{x} \wedge \mathbf{y} := (\min\{[\mathbf{x}]_i, [\mathbf{y}]_i\})_{i=1}^d, \tag{1}$$

respectively. Clearly, we have $\mathbf{x} \wedge \mathbf{y} \leq \mathbf{x} \leq \mathbf{x} \vee \mathbf{y}$. We use $\| \cdot \|$ to denote the Euclidean norm, and $\| \cdot \|_\infty$ to denote the supremum norm. In the paper, we consider a bounded convex domain $\mathcal{K}$ and w.l.o.g. assume that $\mathcal{K} \subseteq [0,1]^d$. We say that $\mathcal{K}$ is *down-closed* (d.c.) if there is a point $\mathbf{u} \in \mathcal{K}$ such that for all $\mathbf{z} \in \mathcal{K}$, we have $\{\mathbf{x} \mid \mathbf{u} \leq \mathbf{x} \leq z\} \subseteq \mathcal{K}$. The *diameter* $D$ of the convex domain $\mathcal{K}$ is defined as $D := \sup_{\mathbf{x}, \mathbf{y} \in \mathcal{K}} \|\mathbf{x} - \mathbf{y}\|$. We use $\mathbb{B}_r(x)$ to denote the open ball of radius $r$ centered at $\mathbf{x}$. More generally, for a subset $X \subseteq \mathbb{R}^d$, we define $\mathbb{B}_r(X) := \bigcup_{x \in X} \mathbb{B}_r(x)$. If $A$ is an affine subspace of $\mathbb{R}^d$, then we define $\mathbb{B}_r^A(X) := A \cap \mathbb{B}_r(X)$. For a function $F : \mathcal{D} \to \mathbb{R}$ and a set $\mathcal{L}$, we use $F|_\mathcal{L}$ to denote the restriction of $F$ to the set $\mathcal{D} \cap \mathcal{L}$. For a linear space $\mathcal{L}_0 \subseteq \mathbb{R}^d$, we use $P_{\mathcal{L}_0} : \mathbb{R}^d \to \mathcal{L}_0$ to denote the projection onto $\mathcal{L}_0$. We will use $\mathbb{R}_+^d$ to denote the set $\{\mathbf{x} \in \mathbb{R}^d | \mathbf{x} \geq 0\}$. For any set $X \subseteq \mathbb{R}^d$, the affine hull of $X$, denoted by $\mathrm{aff}(X)$, is defined to be the intersection of all affine subsets of $\mathbb{R}^d$ that contain $X$. The *relative interior* of a set $X$ is defined by

$$\mathrm{relint}(X) := \{\mathbf{x} \in X \mid \exists \varepsilon > 0, \mathbb{B}_\varepsilon^{\mathrm{aff}(X)}(\mathbf{x}) \subseteq X\}.$$

It is well known that for any non-empty convex set $\mathcal{K}$, the set $\mathrm{relint}(\mathcal{K})$ is always non-empty. We will always assume that the feasible set contains at least two points and therefore $\dim(\mathrm{aff}(\mathcal{K})) \geq 1$, otherwise the optimization problem is trivial and there is nothing to solve.

A set function $f : \{0,1\}^d \to \mathbb{R}^+$ is called *submodular* if for all $\mathbf{x}, \mathbf{y} \in \{0,1\}^d$ with $\mathbf{x} \geq \mathbf{y}$, we have

$$f(\mathbf{x} \vee \mathbf{a}) - f(\mathbf{x}) \leq f(\mathbf{y} \vee \mathbf{a}) - f(\mathbf{y}), \qquad \forall \mathbf{a} \in \{0,1\}^d. \tag{2}$$

Submodular functions can be generalized over continuous domains. A function $F : [0,1]^d \to \mathbb{R}^+$ is called *DR-submodular* if for all vectors $\mathbf{x}, \mathbf{y} \in [0,1]^d$ with $\mathbf{x} \leq \mathbf{y}$, any basis vector $\mathbf{e}_i = (0, \cdots, 0, 1, 0, \cdots, 0)$ and any constant $c > 0$ such that $\mathbf{x} + c\mathbf{e}_i \in [0,1]^d$ and $\mathbf{y} + c\mathbf{e}_i \in [0,1]^d$, it holds that

$$F(\mathbf{x} + c\mathbf{e}_i) - F(\mathbf{x}) \geq F(\mathbf{y} + c\mathbf{e}_i) - F(\mathbf{y}). \tag{3}$$

Note that if function $F$ is differentiable then the diminishing-return (DR) property (3) is equivalent to $\nabla F(\mathbf{x}) \geq \nabla F(\mathbf{y})$ for $\mathbf{x} \leq \mathbf{y}$ with $\mathbf{x}, \mathbf{y} \in [0,1]^d$. A function $F : \mathcal{D} \to \mathbb{R}^+$ is *G-Lipschitz continuous* if for all $\mathbf{x}, \mathbf{y} \in \mathcal{D}$, $\|F(\mathbf{x}) - F(\mathbf{y})\| \leq G\|\mathbf{x} - \mathbf{y}\|$. A differentiable function $F : \mathcal{D} \to \mathbb{R}^+$ is *L-smooth* if for all $\mathbf{x}, \mathbf{y} \in \mathcal{D}$, $\|\nabla F(\mathbf{x}) - \nabla F(\mathbf{y})\| \leq L\|\mathbf{x} - \mathbf{y}\|$.

A (possibly randomized) offline algorithm is said to be an $\alpha$-approximation algorithm (for constant $\alpha \in (0,1]$) with $\epsilon \geq 0$ additive error for a class of maximization problems over non-negative functions if, for any problem instance $\max_{\mathbf{z} \in \mathcal{K}} F(\mathbf{z})$, the algorithm output $\mathbf{x}$ that satisfies the following relation with the optimal solution $\mathbf{z}^*$

$$\alpha F(\mathbf{z}^*) - \mathbb{E}[F(\mathbf{x})] \leq \epsilon, \tag{4}$$

where the expectation is with respect to the (possible) randomness of the algorithm. Further, we assume an oracle that can query the value $F(\mathbf{x})$ or the gradient $\nabla F(\mathbf{x})$. The number of calls to the oracle to achieve the error in (4) is called the *evaluation complexity*.

# 3 Offline Algorithms and Guarantees

In this section, we consider the problem of maximizing a DR-submodular function over a general convex set in sixteen different cases, enumerated in Table 1. After setting up the problem in Section 3.1, we then explain two key elements of our proposed algorithm when we only have access to a value oracle, (i) the Black Box Gradient Estimate (BBGE) procedure (Algorithm 1) to balance bias and variance in estimating gradients (Section 3.2) and (ii) the construction of a shrunken feasible region to avoid infeasible value oracle queries during the BBGE procedure (Section 3.3). Our main algorithm is proposed in Section 3.4 and analyzed in Section 3.5.

## 3.1 Problem Setup

We consider a general *non-oblivious* constrained stochastic optimization problem

$$\max_{\mathbf{z} \in \mathcal{K}} F(\mathbf{z}) := \max_{\mathbf{z} \in \mathcal{K}} \mathbb{E}_{\mathbf{x} \sim p(\mathbf{x};\mathbf{z})}[\hat{F}(\mathbf{z}, \mathbf{x})], \tag{5}$$

where $F$ is a DR-submodular function, and $\hat{F} : \mathcal{K} \times \mathfrak{X} \to \mathbb{R}$ is determined by $\mathbf{z}$ and the random variable $\mathbf{x}$ which is independently sampled according to $\mathbf{x} \sim p(\mathbf{x};\mathbf{z})$. We say the oracle has variance $\sigma^2$ if $\sup_{\mathbf{z} \in \mathcal{K}} \mathrm{var}_{\mathbf{x} \sim p(\mathbf{x};\mathbf{z})}[\hat{F}(\mathbf{z}, \mathbf{x})] = \sigma^2$. In particular, when $\sigma = 0$, then we say we have access to an exact (deterministic) value oracle. We will use $\hat{F}(\mathbf{z})$ to denote the random variables $\hat{F}(\mathbf{z}, \mathbf{x})$ where $\mathbf{x}$ is a random variable with distribution $p(;\mathbf{z})$. Similarly, we say we have access to a stochastic gradient oracle if we can sample from function $\hat{G} : \mathcal{K} \times \mathfrak{Y} \to \mathbb{R}$ such that $\nabla F(\mathbf{z}) = \mathbb{E}_{\mathbf{y} \sim q(\mathbf{y};\mathbf{z})}[\hat{G}(\mathbf{z}, \mathbf{y})]$, and $\hat{G}$ is determined by $\mathbf{z}$ and the random variable $\mathbf{y}$ which is sampled according to $\mathbf{y} \sim q(\mathbf{y};\mathbf{z})$. Note that oracles are only defined on the feasible set. We will use $\hat{G}(\mathbf{z})$ to denote the random variables $\hat{G}(\mathbf{z}, \mathbf{y})$ where $\mathbf{y}$ is a random variable with distribution $q(;\mathbf{z})$.

**Assumption 1.** We assume that $F : [0, 1]^d \to \mathbb{R}$ is DR-submodular, first-order differentiable, non-negative, $G$-Lipschitz for some $G < \infty$, and $L$-smooth for some $L < \infty$. We also assume the feasible region $\mathcal{K}$ is a closed convex domain in $[0, 1]^d$ with at least two points. Moreover, we also assume that we either have access to a value oracle with variance $\sigma_0^2 \geq 0$ or a gradient oracle with variance $\sigma_1^2 \geq 0$.

*Remark* 1. The proposed algorithm does not need to know the values of $L$, $G$, $\sigma_0$ or $\sigma_1$. However, these constants appear in the final expressions of the number of oracle calls and the regret bounds.

## 3.2 Black Box Gradient Estimate

Without access to a gradient oracle (i.e., first-order information), we estimate gradient information using samples from a value oracle. We will use a variation of the "smoothing trick" technique [14, 17, 1, 27, 31, 8, 33], which involves averaging through spherical sampling around a given point.

**Definition 1** (Smoothing Trick). For a function $F : \mathcal{D} \to \mathbb{R}$ defined on $\mathcal{D} \subseteq \mathbb{R}^d$, its $\delta$-smoothed version $\tilde{F}_\delta$ is given as

$$\tilde{F}_\delta(\mathbf{x}) := \mathbb{E}_{\mathbf{z} \sim \mathbb{B}_\delta^{\mathrm{aff}(\mathcal{D})}(\mathbf{x})}[F(\mathbf{z})] = \mathbb{E}_{\mathbf{v} \sim \mathbb{B}_1^{\mathrm{aff}(\mathcal{D})-\mathbf{x}}(0)}[F(\mathbf{x} + \delta \mathbf{v})], \tag{6}$$

where $\mathbf{v}$ is chosen uniformly at random from the $\dim(\mathrm{aff}(\mathcal{D}))$-dimensional ball $\mathbb{B}_1^{\mathrm{aff}(\mathcal{D})-\mathbf{x}}(0)$. Thus, the function value $\tilde{F}_\delta(\mathbf{x})$ is obtained by "averaging" $F$ over a sliced ball of radius $\delta$ around $\mathbf{x}$.

When the value of $\delta$ is clear from the context, we may drop the subscript and simply use $\tilde{F}$ to denote the smoothed version of $F$. It can be easily seen that if $F$ is DR-submodular, $G$-Lipschitz continuous, and $L$-smooth, then so is $\tilde{F}$ and $\|\tilde{F}(\mathbf{x}) - F(\mathbf{x})\| \leq \delta G$, for any point in the domain of both functions. Moreover, if $F$ is monotone, then so is $\tilde{F}$ (Lemma 3). Therefore $\tilde{F}_\delta$ is an approximation of the function $F$. A maximizer of $\tilde{F}_\delta$ also maximizes $F$ approximately.

Our definition of smoothing trick differs from the standard usage by accounting for the affine hull containing $\mathcal{D}$. This will be of particular importance when the feasible region is of (affine) dimension less than $d$, such as when there are equality constraints. When $\mathrm{aff}(\mathcal{D}) = \mathbb{R}^d$, our definition reduces to the standard definition of the smoothing trick. In this case, it is well-known that the gradient of the smoothed function $\tilde{F}_\delta$ admits an unbiased one-point estimator [14, 17]. Using a two-point estimator instead of the one-point estimator results in smaller variance [1, 27]. In Algorithm 1, we adapt the two-point estimator to the general setting.

## 3.3 Construction of $\mathcal{K}_\delta$

We want to run Algorithm 1 as a subroutine within the main algorithm to estimate the gradient. However, in order to run Algorithm 1, we need to be able to query the oracle within the set $\mathbb{B}_\delta^{\mathrm{aff}(\mathcal{K})}(\mathbf{x})$. Since the oracle can only be queried at points within the feasible set, we need to restrict our attention to a set $\mathcal{K}_\delta$ such that $\mathbb{B}_\delta^{\mathrm{aff}(\mathcal{K})}(\mathcal{K}_\delta) \subseteq \mathcal{K}$. On the other hand, we want the optimal point of $F$ within $\mathcal{K}_\delta$ to be close to the optimal point of $F$ within $\mathcal{K}$. One way to ensure

---

**Algorithm 1** Black Box Gradient Estimate (BBGE)

1: **Input:** Point $\mathbf{z}$, sampling radius $\delta$, constraint linear space $\mathcal{L}_0$, $k = \dim(\mathcal{L}_0)$, batch size $B$
2: Sample $\mathbf{u}_1, \cdots, \mathbf{u}_B$ i.i.d. from $S^{d-1} \cap \mathcal{L}_0$
3: For $i = 1$ to $B$, let $\mathbf{y}_i^+ \leftarrow \mathbf{z} + \delta \mathbf{u}_i, y_i^- \leftarrow \mathbf{z} - \delta \mathbf{u}_i$, and evaluate $\hat{F}(\mathbf{y}_i^+), \hat{F}(\mathbf{y}_i^-)$
4: $\mathbf{g} \leftarrow \frac{1}{B} \sum_{i=1}^B \frac{k}{2\delta} \left[ \hat{F}(\mathbf{y}_i^+) - \hat{F}(\mathbf{y}_i^-) \right] \mathbf{u}_i$
5: Output $\mathbf{g}$

---

that is to have $\mathcal{K}_\delta$ not be too small. More formally, we want that $\mathbb{B}_{\delta'}^{\mathrm{aff}(\mathcal{K})}(\mathcal{K}_\delta) \supseteq \mathcal{K}$, for some value of $\delta' \geq \delta$ that is not too large. The constraint boundary could have a complex geometry, and simply maintaining a $\delta$ sized margin away from the boundary can result in big gaps between the boundary of $\mathcal{K}$ and $\mathcal{K}_\delta$. For example, in two dimensions, if $\mathcal{K}$ is polyhedral and has an acute angle, maintaining a $\delta$ margin away from both edges adjacent to the acute angle means the closest point in the $\mathcal{K}_\delta$ to the corner may be much more than $\delta$. For this construction, we choose a $\mathbf{c} \in \mathrm{relint}(\mathcal{K})$ and a real number $r > 0$ such that $\mathbb{B}_r^{\mathrm{aff}(\mathcal{K})}(\mathbf{c}) \subseteq \mathcal{K}$. For any $\delta < r$, we define

$$\mathcal{K}_\delta^{\mathbf{c},r} := (1 - \frac{\delta}{r})\mathcal{K} + \frac{\delta}{r}\mathbf{c}. \tag{7}$$

Clearly if $\mathcal{K}$ is downward-closed, then so is $\mathcal{K}_\delta^{\mathbf{c},r}$. Lemma 7 shows that for any such choice of $\mathbf{c}$ and $r > 0$, we have $\frac{\delta'}{\delta} \leq \frac{D}{r}$. See Appendix G for more details about the choice of $\mathbf{c}$ and $r$. We will drop the superscripts in the rest of the paper when there is no ambiguity.

*Remark* 2. This construction is similar to the one carried out in [31] which was for $d$-dimensional downward-closed sets. Here we impose no restrictions on $\mathcal{K}$ beyond Assumption 1. A simpler construction of shrunken constraint set was proposed in [8]. However, as we discuss in Appendix D, they require to be able to query outside of the constraint set.

## 3.4 Generalized DR-Submodular Frank-Wolfe

The pseudocode of our proposed offline algorithm, Generalized DR-Submodular Frank-Wolfe, is shown in Algorithm 2. At a high-level, it follows the basic template of Frank-Wolfe type methods, where over the course of a pre-specified number of iterations, the gradient (or a surrogate thereof) is calculated, an optimization sub-routine with a linear objective is solved to find a feasible point whose difference (with respect to the current solution) has the largest inner product with respect to the gradient, and then the current solution is updated to move in the direction of that feasible point.

However, there are a number of important modifications to handle properties of the objective function, constraint set, and ora-

---

**Algorithm 2** Generalized DR-Submodular Frank-Wolfe

1: **Input:** Constraint set $\mathcal{K}$, iteration limit $N \geq 4$, sampling radius $\delta$, gradient step-size $\{\rho_n\}_{n=1}^N$
2: Construct $\mathcal{K}_\delta$
3: Pick any $\mathbf{z}_1 \in \mathrm{argmin}_{\mathbf{z} \in \mathcal{K}_\delta} \|\mathbf{z}\|_\infty$
4: $\bar{\mathbf{g}}_0 \leftarrow \mathbf{0}$
5: **for** $n = 1$ **to** $N$ **do**
6: $\quad \mathbf{g}_n \leftarrow \text{estimate-grad}(\mathbf{z}_n, \delta, \mathcal{L}_0 = \mathrm{aff}(\mathcal{K}) - \mathbf{z}_1)$
7: $\quad \bar{\mathbf{g}}_n \leftarrow (1 - \rho_n)\bar{\mathbf{g}}_{n-1} + \rho_n \mathbf{g}_n$
8: $\quad \mathbf{v}_n \leftarrow \text{optimal-direction}(\bar{\mathbf{g}}_n, \mathbf{z}_n)$
9: $\quad \mathbf{z}_{n+1} \leftarrow \text{update}(\mathbf{z}_n, \mathbf{v}_n, \varepsilon)$
10: **end for**
11: Output $\mathbf{z}_{N+1}$

---

cle type. For the oracle type, for instance, standard Frank-Wolfe methods assume access to a deterministic gradient oracle. Frank-Wolfe methods are known to be sensitive to errors in estimates of the gradient (e.g., see [16]). Thus, when only a stochastic gradient oracle or even more challenging, only a stochastic value oracle is available, the gradient estimators must be carefully designed to balance query complexity on the one hand and output error on the other. The Black Box Gradient Estimate (BBGE) sub-routine, presented in Algorithm 1, utilizes spherical sampling to produce an unbiased gradient estimate. This estimate is then combined with past estimates using momentum, as seen in [23], to control and reduce variance.

Our algorithm design is influenced by state-of-the-art methods that have been developed for specific settings. One of the most closely related works is [8], which also dealt with using value oracle access for optimizing monotone functions. They used momentum and spherical sampling techniques that are similar to the ones we used in our Algorithm 1. However, we modified the sampling procedure and the solution update step. In their work, [8] also considered a shrunken feasible region to avoid sampling close to the boundary. However, they assumed that the value oracle could be queried outside the feasible set (see Appendix D for details).

In Algorithm 3, we consider the following cases for the function and the feasible set.

(A) If $F$ is monotone DR-submodular and $\mathbf{0} \in \mathcal{K}$, we choose

$$\text{optimal-direction}(\bar{\mathbf{g}}_n, \mathbf{z}_n) = \text{argmax}_{\mathbf{v} \in \mathcal{K}_\delta - \mathbf{z}_1} \langle \mathbf{v}, \bar{\mathbf{g}}_n \rangle, \ \text{update}(\mathbf{z}_n, \mathbf{v}_n, \varepsilon) = \mathbf{z}_n + \varepsilon \mathbf{v}_n,$$

and $\varepsilon = 1/N$. We start at a point near the origin and always move to points that are bigger with respect to the partial order on $\mathbb{R}^d$. In this case, since the function is monotone, the optimal direction is a maximal point with respect to the partial order. The choice of $\varepsilon = 1/N$ guarantees that after $N$ steps, we arrive at a convex combination of points in the feasible set and therefore the final point is also in the feasible set. The fact that the origin is also in the feasible set shows that the intermediate points also belong to the feasible set.

(B) If $F$ is non-monotone DR-submodular and $\mathcal{K}$ is a downward closed set containing 0, we choose

$$\text{optimal-direction}(\bar{\mathbf{g}}_n, \mathbf{z}_n) = \text{argmax}_{\substack{\mathbf{v} \in \mathcal{K}_\delta - \mathbf{z}_1 \\ \mathbf{v} \leq \mathbf{1} - \mathbf{z}_n}} \langle \mathbf{v}, \bar{\mathbf{g}}_n \rangle, \ \text{update}(\mathbf{z}_n, \mathbf{v}_n, \varepsilon) = \mathbf{z}_n + \varepsilon \mathbf{v}_n,$$

and $\varepsilon = 1/N$. This case is similar to (A). However, since $F$ is not monotone, we need to choose the optimal direction more conservatively.

(C) If $F$ is monotone DR-submodular and $\mathcal{K}$ is a general convex set, we choose

$$\text{optimal-direction}(\bar{\mathbf{g}}_n, \mathbf{z}_n) = \text{argmax}_{\mathbf{v} \in \mathcal{K}_\delta} \langle \mathbf{v}, \bar{\mathbf{g}}_n \rangle, \ \text{update}(\mathbf{z}_n, \mathbf{v}_n, \varepsilon) = (1 - \varepsilon)\mathbf{z}_n + \varepsilon \mathbf{v}_n,$$

and $\varepsilon = \log(N)/2N$. In this case, if we update like in cases (A) and (B), we do not have any guarantees of ending up in the feasible set, so we choose the update function to be a convex combination. Unlike (B), we do not need to limit ourselves in choosing the optimal direction and we simply choose $\varepsilon$ to obtain the best approximation coefficient.

(D) If $F$ is non-monotone DR-submodular and $\mathcal{K}$ is a general convex set, we choose

$$\text{optimal-direction}(\bar{\mathbf{g}}_n, \mathbf{z}_n) = \text{argmax}_{\mathbf{v} \in \mathcal{K}_\delta} \langle \mathbf{v}, \bar{\mathbf{g}}_n \rangle, \quad \text{update}(\mathbf{z}_n, \mathbf{v}_n, \varepsilon) = (1 - \varepsilon)\mathbf{z}_n + \varepsilon \mathbf{v}_n,$$

and $\varepsilon = \log(2)/N$. This case is similar to (C) and we choose $\varepsilon$ to obtain the best approximation coefficient.

The choice of subroutine estimate-grad and $\rho_n$ depend on the oracle. If we have access to a gradient oracle $\hat{G}$, we set estimate-grad$(\mathbf{z}, \delta, \mathcal{L}_0)$ to be the average of $B$ evaluations of $P_{\mathcal{L}_0}(\hat{G}(\mathbf{z}))$. Otherwise, we run Algorithm 1 with input $\mathbf{z}, \delta, \mathcal{L}_0$. If we have access to a deterministic gradient oracle, then there is no need to use any momentum and we set $\rho_n = 1$. In other cases, we choose $\rho_n = \frac{2}{(n+3)^{2/3}}$.

### 3.5 Approximation Guarantees for the Proposed Offline Algorithm

**Theorem 1.** *Suppose Assumption 1 holds. Let $N \geq 4$, $B \geq 1$ and choose $\mathbf{c} \in \mathcal{K}$ and $r > 0$ according to Section 3.3. If we have access to a gradient oracle, we choose $\delta = 0$, otherwise we choose $\delta \in (0, r/2)$. Then the following results hold for the output $\mathbf{z}_{N+1}$ of Algorithm 2.*

*(A) If $F$ is monotone DR-submodular and $\mathbf{0} \in \mathcal{K}$, then*

$$(1 - e^{-1})F(\mathbf{z}^*) - \mathbb{E}[F(\mathbf{z}_{N+1})] \leq \frac{3DQ^{1/2}}{N^{1/3}} + \frac{LD^2}{2N} + \delta G(2 + \frac{\sqrt{d} + D}{r}). \tag{8}$$

*(B) If $F$ is non-monotone DR-submodular and $\mathcal{K}$ is a downward closed set containing $\mathbf{0}$, then*

$$e^{-1}F(\mathbf{z}^*) - \mathbb{E}[F(\mathbf{z}_{N+1})] \leq \frac{3DQ^{1/2}}{N^{1/3}} + \frac{LD^2}{2N} + \delta G(2 + \frac{\sqrt{d} + 2D}{r}). \tag{9}$$

*(C) If F is monotone DR-submodular and $\mathcal{K}$ is a general convex set, then*

$$\frac{1}{2}F(\mathbf{z}^*) - \mathbb{E}[F(\mathbf{z}_{N+1})] \leq \frac{3DQ^{1/2}\log(N)}{2N^{1/3}} + \frac{4DG + LD^2\log(N)^2}{8N} + \delta G(2 + \frac{D}{r}). \quad (10)$$

*(D) If F is non-monotone DR-submodular and $\mathcal{K}$ is a general convex set, then*

$$\frac{1}{4}(1 - \|\mathbf{z}_1\|_\infty)F(\mathbf{z}^*) - \mathbb{E}[F(\mathbf{z}_{N+1})] \leq \frac{3DQ^{1/2}}{N^{1/3}} + \frac{DG + 2LD^2}{4N} + \delta G(2 + \frac{D}{r}). \quad (11)$$

*In all these cases, we have*

$$Q = \begin{cases} 0 & \textit{det. grad. oracle,} \\ \max\{4^{2/3}G^2, 6L^2D^2 + \frac{4\sigma_1^2}{B}\} & \textit{stoch. grad. oracle with variance } \sigma_1^2 > 0, \\ \max\{4^{2/3}G^2, 6L^2D^2 + \frac{4CkG^2 + 2k^2\sigma_0^2/\delta^2}{B}\} & \textit{value oracle with variance } \sigma_0^2 \geq 0, \end{cases}$$

*C is a constant, $k = \dim(\mathcal{K})$, $D = \operatorname{diam}(\mathcal{K})$, and $\mathbf{z}^*$ is the global maximizer of F on $\mathcal{K}$.*

Theorem 1 characterizes the worst-case approximation ratio $\alpha$ and additive error bounds for different properties of the function and feasible region, where the additive error bounds depend on selected parameters $N$ for the number of iterations, batch size $B$, and sampling radius $\delta$.

The proof of Parts (A)-(D) is provided in Appendix I-L, respectively.

The proof of parts (A), (B) and (D), when we have access to an exact gradient oracle is similar to the proofs presented in [4, 3, 24], respectively. Part (C) is the first analysis of a Frank-Wolfe type algorithm over general convex sets when the oracle can only be queried within the feasible set. When we have access to a stochastic gradient oracle, directly using a gradient sample can result in arbitrary bad performance as shown in Appendix B of [16]. The momentum technique, first used in continuous submodular maximization in [23], is used when we have access to a stochastic gradient oracle. The control on the estimate of the gradient is deferred to Lemma 9. Since the momentum technique is robust to noise in the gradient, when we only have access to a value oracle, we can use Algorithm 1, similar to [8], to obtain an unbiased estimate of the gradient and complete the proof.

Theorem 2 converts those bounds to characterize the oracle complexity for a user-specified additive error tolerance $\epsilon$ based on oracle properties (deterministic/stochastic gradient/value). The 16 combinations of the problem settings listed in Table 1 are enumerated by four cases (A)–(D) in Theorem 1 of function and feasible region properties (resulting in different approximation ratios) and the four cases 1–4 enumerated in Theorem 2 below of oracle properties. For the oracle properties, we consider the four cases as (Case 1): deterministic gradient oracle, (Case 2): stochastic gradient oracle, (Case 3): deterministic value oracle, and (Case 4): stochastic value oracle.

**Theorem 2.** *The number of oracle calls for different oracles to achieve an $\alpha$-approximation error of smaller than $\epsilon$ using Algorithm 1 is*

$$\textbf{Case 1: } \tilde{O}(1/\epsilon), \quad \textbf{Cases 2, 3: } \tilde{O}(1/\epsilon^3), \quad \textbf{Case 4: } \tilde{O}(1/\epsilon^5).$$

*Moreover, in all of the cases above, if F is non-monotone or $0 \in \mathcal{K}$, we may replace $\tilde{O}$ with $O$.*

See Appendix M for proof.

## 4 Online DR-submodular optimization under bandit or semi-bandit feedback

In this section, we first describe the Black-box Explore-Then-Commit algorithm that uses the offline algorithm for exploration, and uses the solution of the offline algorithm for exploitation. This is followed by the regret analysis of the proposed algorithm. This is the first algorithm for stochastic continuous DR-submodular maximization under bandit feedback and obtains state-of-the-art for semi-bandit feedback.

### 4.1 Problem Setup

There are typically two settings considered in online optimization with bandit feedback. The first is the adversarial setting, where the environment chooses a sequence of functions $F_1, \cdots, F_N$ and in each iteration $n$, the agent chooses a point $\mathbf{z}_n$ in the feasible set $\mathcal{K}$, observes $F_n(z_n)$ and receives the reward $F_n(\mathbf{z}_n)$. The goal is to choose the sequence of actions that minimize the following notion of expected $\alpha$-regret.

$$\mathcal{R}_{\mathrm{adv}} := \alpha \max_{\mathbf{z} \in \mathcal{K}} \sum_{n=1}^{N} F_n(\mathbf{z}) - \mathbb{E}\left[\sum_{n=1}^{N} F_n(\mathbf{z}_n)\right]. \tag{12}$$

In other words, the agent's cumulative reward is being compared to $\alpha$ times the reward of the best *constant* action in hindsight. Note that, in this case, the randomness is over the actions of the policy.

The second is the stochastic setting, where the environment chooses a function $F : \mathcal{K} \to \mathbb{R}$ and a stochastic value oracle $\hat{F}$. In each iteration $n$, the agent chooses a point $\mathbf{z}_n$ in the feasible set $\mathcal{K}$, receives the reward $(\hat{F}(\mathbf{z}_n))_n$ by querying the oracle at $z_n$ and observes this reward. Here the outer subscript $n$ indicates that the result of querying the oracle at time $n$, since the oracle is stochastic. The goal is to choose the sequence of actions that minimize the following notion of expected $\alpha$-regret.

$$\mathcal{R}_{\mathrm{stoch}} := \alpha N \max_{\mathbf{z} \in \mathcal{K}} F(\mathbf{z}) - \mathbb{E}\left[\sum_{n=1}^{N} (\hat{F}(\mathbf{z}_n))_n\right] = \alpha N \max_{\mathbf{z} \in \mathcal{K}} F(\mathbf{z}) - \mathbb{E}\left[\sum_{n=1}^{N} F(\mathbf{z}_n)\right] \tag{13}$$

Further, two feedback models are considered – bandit and semi-bandit feedback. In the bandit feedback setting, the agent only observes the value of the function $F_n$ at the point $\mathbf{z}_n$. In the semi-bandit setting, the agent has access to a gradient oracle instead of a value oracle and observes $\hat{G}(\mathbf{z}_n)$ at the point $\mathbf{z}_n$, where $\hat{G}$ is an unbiased estimator of $\nabla F$.

In unstructured multi-armed bandit problems, any regret bound for the adversarial setup could be translated into bounds for the stochastic setup. However, having a non-trivial correlation between the actions of different arms complicates the relation between the stochastic and adversarial settings. Even in linear bandits, the relation between adversarial linear bandits and stochastic linear bandits is not trivial. (e.g. see Section 29 in [19]) While it is intuitively reasonable to assume that the optimal regret bounds for the stochastic case are better than that of the adversarial case, such a result is not yet proven for DR-submodular functions. Thus, while the cases of bandit feedback has been studied in the adversarial setup, the results do not reduce to stochastic setup. We also note that in the cases where there are adversarial setup results, this paper finds that the results in the stochastic setup achieve improved regret bounds (See Table 3 in Supplementary for the comparison).

### 4.2 Algorithm for DR-submodular maximization with Bandit Feedback

The proposed algorithm is described in Algorithm 3. In Algorithm 3, if there is semi-bandit feedback in the form of a stochastic gradient sample for each action $\mathbf{z}_n$, we run the offline algorithm (Algorithm 2) with parameters from the proof of case 2 of Theorem 2 for $T_0 = \lceil T^{3/4} \rceil$ total queries. If only the stochastic reward for each action $\mathbf{z}_n$ is available (bandit feedback), we run the offline algorithm (Algorithm 2) with parameters from the proof of case 4 of Theorem 2 for $T_0 = \lceil T^{5/6} \rceil$ total queries. Then, for the remaining time (exploitation phase), we run the last action in the exploration phase.

---

**Algorithm 3** DR-Submodular Explore-Then-Commit

1: **Input: Horizon $T$, inner time horizon $T_0$**
2: Run Algorithm 2 for $T_0$, with according to parameters described in Theorem 2.
3: **for** remaining time **do**
4:     Repeat the last action of Algorithm 2.
5: **end for**

---

### 4.3 Regret Analysis for DR-submodular maximization with Bandit Feedback

In this section, we provide the regret analysis for the proposed algorithm. We note that by Theorem 2, Algorithm 2 requires a sample complexity of $\tilde{O}(1/\epsilon^5)$ with a stochastic value oracle for

offline problems (any of (A)–(D) in Theorem 1). Thus, the parameters and the results with bandit feedback are the same for all the four setups (A)–(D). Likewise, when a stochastic gradient oracle is available, Algorithm 2 requires a sample complexity of $\tilde{O}(1/\epsilon^3)$. Based on these sample complexities, the overall regret of online DR-submodular maximization problem is given as follows.

**Theorem 3.** *For an online constrained DR-submodular maximization problem over a horizon $T$, where the expected reward function $F$, feasible region type $\mathcal{K}$, and approximation ratio $\alpha$ correspond to any of the four cases (A)–(D) in Theorem 1, Algorithm 3 achieves $\alpha$-regret (13) that is upper-bounded as:*

$$\text{Semi-bandit Feedback (Case 2): } \tilde{O}(T^{3/4}), \qquad \text{Bandit Feedback (Case 4): } \tilde{O}(T^{5/6}).$$

*Moreover, in either type of feedback, if $F$ is non-monotone or $\mathbf{0} \in \mathcal{K}$, we may replace $\tilde{O}$ with $O$.*

See Appendix N for the proof.

## 5  Conclusion

This work provides a novel and unified approach for maximizing continuous DR-submodular functions across various assumptions on function, constraint set, and oracle access types. The proposed Frank-Wolfe based algorithm improves upon existing results for nine out of the sixteen cases considered, and presents new results for offline DR-submodular maximization with only a value oracle. Moreover, this work presents the first regret analysis with bandit feedback for stochastic DR-submodular maximization, covering both monotone and non-monotone functions. These contributions significantly advance the field of DR-submodular optimization (with multiple applications) and open up new avenues for future research in this area.

**Limitations:**  While the number of function evaluations in the different setups considered in the paper are state of the art, lower bounds have not been investigated.

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
