# A   Details of Related Works

## A.1   Offline DR-submodular maximization

The authors of [4] considered the problem of maximizing a monotone DR-submodular function over a downward-closed convex set given a deterministic gradient oracle. They showed that a variant of the Frank-Wolfe algorithm guarantees an optimal $(1 - \frac{1}{e})$-approximation for this problem. While they only claimed their result for downward-closed convex sets, their result holds under a more general setting where the convex set contains the origin. In [3], a non-monotone variant of the algorithm for downward-closed convex sets with $\frac{1}{e}$-approximation was proposed.

The authors of [16] used gradient ascent to obtain $\frac{1}{2}$-guarantees for the maximization of a monotone DR-submodular function over a general convex set given a gradient oracle which could be stochastic. They proved that gradient ascent cannot guarantee better than a $\frac{1}{2}$-approximation by constructing a convex set $\mathcal{K}$ and a function $F : \mathcal{K} \to \mathbb{R}$ such that $F$ has a local maximum that is a $\frac{1}{2}$-approximation of its optimal value on $\mathcal{K}$. They also showed that a Frank-Wolfe type algorithm similar to [4] cannot be directly used when we only have access to a stochastic gradient oracle. [32] extended projected gradient ascent using a line integral method, referred to as boosting, to obtain $(1 - 1/e)$-approximation for convex sets containing the origin. Later, [23] resolved the issue of stochastic gradient oracles with a momentum technique and obtained $(1 - \frac{1}{e})$-approximation in the case of monotone functions over sets that contain the origin, and $\frac{1}{e}$-approximation in the case of non-monotone functions over downward closed sets. In [32] and the first case in [23], while they consider monotone DR-submodular functions over general convex sets $\mathcal{K}$, they query the oracle over the convex hull of $\mathcal{K} \cup \{\mathbf{0}\}$ (See Appendix B).

For non-monotone maps over general convex sets, no constant approximation ratio can be guaranteed in sub-exponential time due to a hardness result by [29]. However, [12] bypassed this issue by finding an approximation guarantee that depends on the geometry of the convex set. Specifically, they showed that given a deterministic gradient oracle for a non-monotone function over a general convex set $\mathcal{K} \subseteq [0, 1]^d$, their proposed algorithm obtains $\frac{1}{3\sqrt{3}}(1 - h)$-approximation of the optimal value where $h := \min_{\mathbf{z} \in \mathcal{K}} \|\mathbf{z}\|_\infty$. An improved sub-exponential algorithm was proposed by [11] that obtained a $\frac{1}{4}(1 - h)$-approximation guarantees, which is optimal. Later, [10] provided the first polynomial time algorithm for this setting with the same approximation coefficient.

*Remark* 3. In the special case of maximizing a non-monotone continuous DR-submodular over a box, i.e. $[0, 1]^d$, one could discretize the problem and use discrete algorithms to solve the continuous version. The technique has been employed in [3] to obtain a $\frac{1}{3}$-approximation and in [5, 26] to obtain $\frac{1}{2}$-approximations for the optimal value. We have not included these results in Table 1 since using discretization has only been successfully applied to the case where the convex set is a box and can not be directly used in more general settings.

## A.2   Online DR-submodular maximization with bandit feedback

There has been growing interest in online DR-submodular maximization in the recent years [7], [6], [31], [28], [25], [33], [13],[24]. Most of these results are focused on adversarial online full-information feedback. In the adversarial setting, the environment chooses a sequence of functions $F_1, \cdots, F_N$ and in each iteration $n$, the agent chooses a point $z_n$ in the feasible set $\mathcal{K}$, observes $F_n$ and receives the reward $F_n(z_n)$. For the regret bound, the agents reward is being compared to $\alpha$ times the reward of the best *constant* action in hindsight. With full-information feedback, if at each iteration when the agent observes $F_n$, it may be allowed to query the value of $\nabla F_n$ or maybe $F_n$ at any number of arbitrary points within the feasible set. Further, we consider stochastic setting, where the environment chooses a function $F : \mathcal{K} \to \mathbb{R}$ and a sequence of independent noise functions $\eta_n : \mathcal{K} \to \mathbb{R}$ with zero mean. In each iteration $n$, the agent chooses a point $\mathbf{z}_n$ in the feasible set $\mathcal{K}$, receives the reward $(F + \eta_n)(\mathbf{z}_n)$ and observes the reward. For the regret bound, the agents reward is being compared to $\alpha$ times the reward of the best action. Detailed formulation of adversarial and stochastic setups and why adversarial results cannot be reduced to stochastic results is given in Section 4.1. In this paper, we consider two feedback models – bandit feedback where only the (stochastic) reward value is available and semi-bandit feedback where a single stochastic sample of the gradient at the location is provided.

| Function | Set | Setting | Reference | Appx. | Regret |
|---|---|---|---|---|---|
| Monotone | $0 \in \mathcal{K}$ | stoch. | This paper | $1 - 1/e$ | $O(T^{5/6})$ |
| | | adv. | [31] | $1 - 1/e$ | $O(T^{8/9})$ |
| | | | [25] | $1 - 1/e$ | $O(T^{5/6})$ |
| | | | [30]† | $1 - 1/e$ | $O(T^{3/4})$ |
| | general | stoch. | This paper | $1/2$ | $\tilde{O}(T^{5/6})$ |
| | | adv. | - | | |
| Non-monotone | d.c. | stoch. | This paper | $1/e$ | $O(T^{5/6})$ |
| | | adv. | [33] | $1/e$ | $O(T^{8/9})$ |
| | general | stoch. | This paper | $\frac{1-h}{4}$ | $O(T^{5/6})$ |
| | | adv. | - | | |

Table 3: This table presents the different results for the regret for DR-submodular maximization under bandit feedback, and gives the related works and regret bounds in the adversarial case. Note that the result marked by † uses a convex optimization subroutine at each iteration which could be even more computationally expensive than projection. As before, we have $h := \min_{\mathbf{x} \in \mathcal{K}} \|x\|_\infty$.

**Bandit Feedback:** We note that this paper is the first work for bandit feedback for stochastic online DR-submodular maximization. The prior works on this topic has been in the adversarial setup [31, 33, 25, 30], and the results in this work is compared with their results in Table 3. In [31], the adversarial online setting with bandit feedback has been studied for monotone DR-submodular functions over downward-closed convex sets. Later [33] extended this framework to the setting with non-monotone DR-submodular functions over downward-closed convex sets. [25] described a framework for converting certain greedy-type offline algorithms with robustness guarantees into adversarial online algorithms for both full-information and bandit feedback. They apply their framework to obtain algorithms for non-monotone functions over a box, with $\frac{1}{2}$-regret of $\tilde{O}(T^{4/5})$, and monotone function over downward-closed convex sets. The offline algorithm they use for downward-closed convex sets is the one described in [4] which only requires the convex set to contain the origin. They also use the construction of the shrunk constraint set described in [31]. By replacing that construction with ours, the result of [25] could be extended to monotone functions over all convex sets containing the origin. [30] improved the regret bound for monotone functions over convex sets containing the origin to $O(T^{3/4})$. However, they use a convex optimization subroutine at each iteration which could be even more computationally expensive than projection.

**Semi-bandit Feedback:** In semi-bandit feedback, a single stochastic sample of the gradient is available. The problem has been considered in [6], while the results have an error (See Appendix D). Further, they only obtain $\frac{1}{e}$-regret for the monotone case. One could consider a generalization of the adversarial and stochastic setting in the following manner. The environment chooses a sequence of functions $F_n$ and a sequence of value oracles $\hat{F}_n$ such that $\hat{F}_n$ estimates $F_n$. In each iteration $n$, the agent chooses a point $\mathbf{z}_n$ in the feasible set $\mathcal{K}$, receives the reward $(\hat{F}_n(\mathbf{z}_n))_n$ by querying the oracle at $z_n$ and observes this reward. The goal is to choose the sequence of actions that minimize the following notion of expected $\alpha$-regret.

$$
\mathcal{R}_{\text{stoch-adv}} := \alpha \max_{\mathbf{z} \in \mathcal{K}} \sum_{n=1}^{N} F_n(\mathbf{z}) - \mathbb{E}\left[\sum_{n=1}^{N} (\hat{F}_n(\mathbf{z}_n))_n\right]
$$
$$
= \alpha \max_{\mathbf{z} \in \mathcal{K}} \sum_{n=1}^{N} F_n(\mathbf{z}) - \mathbb{E}\left[\sum_{n=1}^{N} F_n(\mathbf{z}_n)\right] \tag{14}
$$

Algorithm 3 in [7] solves this problem in semi-bandit feedback setting with a deterministic value oracle and stochastic gradient oracles. Any bound for a problem in this setting implies bounds for stochastic semi-bandit and adversarial semi-bandit settings. The same is true for Mono-Frank-Wolfe Algorithms in [31, 33]. We have included these results in Table 2 as benchmark to compare with results in stochastic setting.

## B    Constraint Set and Query Set

In this work, we made the assumption that the query set is identical to the constraint set, i.e. oracles can only be queried within the constraint set. To the best of our knowledge, except in the context of online optimization with (semi-)bandit feedback, this is the first work on DR-submodular maximization that explicitly considers this assumption. Previous works assumed that we may query the oracle at any point within the unit box $[0,1]^d$. Algorithms designed for non-monotone functions in prior works already satisfied the assumption we consider, so no changes in algorithms, proofs, or results are needed. However, the situation is different when the function is monotone. This assumption allows us to explain a previously unexplained gap in approximation guarantees for monotone DR-submodular maximization. Specifically, some prior works (enumerated below) studying monotone DR-submodular maximization over general convex sets obtained approximation guarantees of $1/2$ while others obtained $1 - 1/e$.

First we describe how some of previous results in literature with no apparent restriction on the query set may be reformulated as problems where the query set is equal to the constraint set. Let $\mathcal{K} \subseteq [0,1]^d$ be a convex set, and define $\mathcal{K}^*$ as the convex hull of $\mathcal{K} \cup \{\mathbf{0}\}$. For a problem in the setting of monotone functions over a general set $\mathcal{K}$, we can consider the same problem on $\mathcal{K}^*$. Since the function is monotone, the optimal solution in $\mathcal{K}^*$ is the same as the optimal solution in $\mathcal{K}$. However, solving this problem in $\mathcal{K}^*$ may require evaluating the function in the larger set $\mathcal{K}^*$, which may not always be possible. In fact, the result of [23] and [32] mentioned in Table 1 are for monotone functions over general convex sets $\mathcal{K}$, but their algorithms require evaluating the function on $\mathcal{K}^*$. This is why we have classified their results as algorithms for convex sets that contain the origin. The problem of offline DR-submodular maximization with only a value oracle was first considered by [8] for monotone maps over convex sets that contain the origin. However, their result requires querying in a neighborhood of $\mathcal{K}^*$ which violates our requirement to only query the oracle within the feasible set (see Appendix D).

In [16], a $1/2$ approximation guarantee was obtained by a projected gradient ascent method and this was shown by proving that the algorithm tends to a stationary point and proving that any stationary point is at least $1/2$ as good as the optimal point. Moreover, they construct examples with stationary points that are no better that $1/2$ of the optimal point.

The $1 - 1/e$ approximation guarantee was first reported for Frank-Wolfe methods, which (superficially) suggests that the gap may be due to algorithm or analysis differences. Later, [32] developed a projected gradient ascent based method that obtains a $1 - 1/e$ approximation guarantee where they consider general constraint set but their query set contains the origin.

However, the gap is not attributable to algorithm or analysis differences, but instead due to the fact that the query sets are different. In other words, the results that obtain a $1 - 1/e$ approximation guarantee are solving a different problem than the ones obtaining a $1/2$ approximation guarantee. A key ingredient to obtain $1 - 1/e$ is the ability to query the (gradient) oracle within the convex hull of $\mathcal{K} \cup \{0\}$. For monotone submodular maximization over general convex sets (not necessarily containing the origin), we can only guarantee a coefficient of $1/2$, both for Frank-Wolfe type methods (our work) and projection based methods (i.e. [16]). Therefore, the $1/2$ approximation could very well be optimal in its own setting.

To the best of our knowledge, in every paper where the $1/2$ approximation coefficient and $1 - 1/e$ approximation coefficient in the monotone setting are compared, the comparison was (unwittingly) between problems that are inherently mathematically different: [16] and [7] in experiments and main text; [6] and [8] in experiments; [33, 24], and [12] in related work section, [23] in the introduction and Table 2, [32] and [13] in the main claims.

**Conjecture**    *The problem of maximizing a monotone DR-submodular continuous function subject to a general convex constraint, where oracle queries are limited to the feasible region, is NP-hard. For any $\epsilon > 0$, it cannot be approximated in polynomial time to within a ratio of $1/2 + \epsilon$ (up to low-order terms), unless $RP = NP$.*

## C  Brief discussion on oracle models in applications

For many problems, the ability to evaluate gradients directly requires strong assumptions about problem-specific parameters. Influence maximization and profit maximization form a family of problems that model choosing advertising resource allocations to maximize the expected number of customers, where there is an underlying diffusion model for how advertising resources spent (stochastically) activate customers over a social network. For common diffusion models, the objective function is known to be DR-submodular (see for instance [3] or [15]). The revenue (expected number of activated customers) is a monotone objective function; total profit (revenue from activated customers minus advertising costs) is a non-monotone objective. One significant challenge with these problems is that the objective function (and the gradients) cannot be analytically evaluated for general (non-bipartite) networks, even if all the underlying diffusion model parameters are known exactly. The mildest assumptions on knowledge/observability of the network diffusions for offline variants (respectively actions for online variants), especially fitting for user privacy and/or third-party access, leads to instantiations of queries as the agent selecting an advertising allocation within the budget (i.e., feasible point) and observing a (stochastic) count of activated customers. This corresponds to stochastic value oracle queries over the feasible region (respectively bandit feedback for online variants).

## D  Comments on previous results in literature

**Construction of $\mathcal{K}'$ and error estimate in [8]**     In [8], the set $\mathcal{K}' + \delta\mathbf{1}$ plays a role similar to the set $\mathcal{K}_\delta$ defined in this paper. Algorithm 2, in the case with access to value oracle for monotone DR-submodular function with the constraint set $\mathcal{K}$, such that $\mathrm{aff}(\mathcal{K}) = \mathbb{R}^d$ and $\mathbf{0} \in \mathcal{K}$, reduced to BBCG algorithm in [8] if we replace $\mathcal{K}_\delta$ with their construction of $\mathcal{K}' + \delta\mathbf{1}$. In their paper, $\mathcal{K}'$ is defined by

$$\mathcal{K}' := (\mathcal{K} - \delta\mathbf{1}) \cap [0, 1 - 2\delta]^d. \tag{15}$$

There are a few issues with this construction and the subsequent analysis that requires more care.

1. *The BBCG algorithm almost always needs to be able to query the value oracle outside the feasible set.*
   We have

   $$\mathcal{K}' + \delta\mathbf{1} = \mathcal{K} \cap [\delta, 1 - \delta]^d.$$

   The BBCG algorithm starts at $\delta\mathbf{1}$ and behaves similar to Algorithm 2 in the monotone $\mathbf{0} \in \mathcal{K}$ case. It follows that the set of points that BBCG requires to be able to query is

   $$Q_\delta := \mathbb{B}_\delta(\text{convex-hull}((\mathcal{K}' + \delta\mathbf{1}) \cup \{\delta\mathbf{1}\})) = \mathbb{B}_\delta(\text{convex-hull}(\mathcal{K} \cup \{\delta\mathbf{1}\}) \cap [\delta, 1 - \delta]^d).$$

   If $\mathbf{1} \in \mathcal{K}$, then the problem becomes trivial since $F$ is monotone. If $\mathcal{K}$ is contained in the boundary of $[0, 1]^d$, then we need to restrict ourselves to the affine subspace containing $\mathcal{K}$ and solve the problem in a lower dimension in order to be able to use BBCG algorithm as $\mathcal{K}'$ will be empty otherwise. We want to show that in all other cases, $Q_\delta \setminus \mathcal{K} \neq \emptyset$. If $\mathcal{K}'$ is non-empty and $\mathbf{1} \notin \mathcal{K}$, then let $\mathbf{x}_\delta$ be a maximizer of $\|\cdot\|_\infty$ over $\mathcal{K}' + \delta\mathbf{1}$. If $\mathbf{x}_\delta \neq (1 - \delta)\mathbf{1}$, then there is a point $\mathbf{y} \in \mathbb{B}_\delta(\mathbf{x}_\delta) \cap [\delta, 1 - \delta]^d \subseteq Q_\delta$ such that $\mathbf{y} > \mathbf{x}$ which implies that $\mathbf{y} \notin \mathcal{K}$. Therefore, we only need to prove the statement when $(1 - \delta)\mathbf{1} \in \mathcal{K} \cap [\delta, 1 - \delta]^d$ for all small $\delta$. In this case, since $\mathcal{K}$ is closed, we see that $(1 - \delta)\mathbf{1} \to \mathbf{1} \in \mathcal{K}$. In other words, except in trivial cases, BBCG always requires being able to query outside the feasible set.

2. *The exact error bound could be arbitrarily far away from the correct error bound depending on the geometry of the constraint set.*
   In Equation (69) in the appendix of [8], it is mentioned that

   $$\tilde{F}(\mathbf{x}_\delta^*) \geq \tilde{F}(\mathbf{x}^*) - \delta G\sqrt{d}, \tag{16}$$

   where $\mathbf{x}^*$ is the optimal solution and $\mathbf{x}_\delta^*$ is the optimal solution within $\mathcal{K}' + \delta\mathbf{1}$ and $G$ is the Lipschitz constant. Next we construct an example where this inequality does not hold.
   Consider the set $\mathcal{K} = \{(x, y) \in [0, 1]^2 \mid x + \lambda y \leq 1\}$ for some value of $\lambda$ to be specified and let $F((x, y)) = Gx$. Clearly we have $\mathbf{x}^* = (1, 0)$. Thus, for any $\delta > 0$, we have

   $$\mathcal{K}' + \delta\mathbf{1} = \{(x, y) \in [\delta, 1 - \delta]^2 \mid x + \lambda y \leq 1\}.$$

It follows that when $\lambda \leq \frac{1}{\delta} - 1$, then $\mathcal{K}'$ is non-empty and $\mathbf{x}_\delta^* = (1 - \lambda\delta, \delta)$. Then we have

$$\tilde{F}(\mathbf{x}_\delta^*) - \tilde{F}(\mathbf{x}^*) = -\lambda\delta G.$$

Therefore, (16) is correct if and only if $\lambda \leq \sqrt{d} = \sqrt{2}$. Since this does not hold in general as $\lambda$ depends on the geometry of the convex set, this equation is not true in general making the overall proof incorrect. The issue here is that $\lambda$, which depends on the geometry of the convex set $\mathcal{K}$, should appear in (16). Without restricting ourselves to convex sets with "controlled" geometry and without including a term, such as $\frac{1}{r}$ in Theorem 1, we would not be able to use this method to obtain an error bound. We note that while their analysis has an issue, the algorithm is still fine. Using a proof technique similar to ours, their proof can be fixed, more precisely, we can modify (16) in a manner similar to (24) and (31), depending on the case, and that will help fix their proofs.

**One-Shot Frank-Wolfe algorithm in [6]** In [6], the authors claim their proposed algorithm, One-Shot Frank-Wolfe (OSFW), achieves a $(1 - \frac{1}{e})$-regret for monotone DR-submodular maximization under semi-bandit feedback for general convex set with oracle access to the entire domain of $F$, i.e. $[0, 1]^d$. In their regret analysis in the last page of the supplementary material, the inequality $(1 - 1/T)^t \leq 1/e$ is used for all $0 \leq t \leq T - 1$. Such an inequality holds for $t = T$ but as $t$ decreases, the value of $(1 - 1/T)^t$ becomes closer to 1 and the inequality fails. If we do not use this inequality and continue with the proof, we end up with the following approximation coefficient.

$$1 - \frac{1}{T}\sum_{t=0}^{T-1}(1 - 1/T)^t = 1 - \frac{1}{T} \cdot \frac{1 - (1 - 1/T)^T}{1 - (1 - 1/T)} = 1 - (1 - (1 - 1/T)^T) = (1 - 1/T)^T \sim \frac{1}{e}.$$

# E    Useful lemmas

Here we state some lemmas from the literature that we will need in our analysis of DR-submodular functions.

**Lemma 1** (Lemma 2.2 of [24]). *For any two vectors $\mathbf{x}, \mathbf{y} \in [0, 1]^d$ and any continuously differentiable non-negative DR-submodular function $F$ we have*

$$F(\mathbf{x} \vee \mathbf{y}) \geq (1 - \|\mathbf{x}\|_\infty)F(\mathbf{y}).$$

The following lemma can be traced back to [16] (see Inequality 7.5 in the arXiv version), and is also explicitly stated and proved in [12].

**Lemma 2** (Lemma 1 of [12]). *For every two vectors $\mathbf{x}, \mathbf{y} \in [0, 1]^d$ and any continuously differentiable non-negative DR-submodular function $F$ we have*

$$\langle \nabla F(\mathbf{x}), \mathbf{y} - \mathbf{x} \rangle \geq F(\mathbf{x} \vee \mathbf{y}) + F(\mathbf{x} \wedge \mathbf{y}) - 2F(\mathbf{x}).$$

# F    Smoothing trick

The following Lemma is well-known when $\text{aff}(\mathcal{D}) = \mathbb{R}^d$ (e.g., Lemma 1 in [8], Lemma 7 in [31]). The proof in the general case is similar to the special case $\text{aff}(\mathcal{D}) = \mathbb{R}^d$.

**Lemma 3.** *If $F : \mathcal{D} \to \mathbb{R}$ is DR-submodular, $G$-Lipschitz continuous, and $L$-smooth, then so is $\tilde{F}_\delta$ and for any $\mathbf{x} \in \mathcal{D}$ such that $\mathbb{B}_\delta^{\text{aff}(\mathcal{D})}(\mathbf{x}) \subseteq \mathcal{D}$, we have*

$$\|\tilde{F}_\delta(\mathbf{x}) - F(\mathbf{x})\| \leq \delta G.$$

*Moreover, if $F$ is monotone, then so is $\tilde{F}_\delta$.*

*Proof.* Let $A := \text{aff}(\mathcal{D})$ and $A_0 := \text{aff}(\mathcal{D}) - \mathbf{x}$ for some $\mathbf{x} \in \mathcal{D}$. Using the assumption that $F$ is $G$-Lipschitz continuous, we have

$$\begin{aligned}
|\tilde{F}(\mathbf{x}) - \tilde{F}(\mathbf{y})| &= \left|\mathbb{E}_{\mathbf{v} \sim \mathbb{B}_1^{A_0}(\mathbf{0})}[F(\mathbf{x} + \delta\mathbf{v}) - F(\mathbf{y} + \delta\mathbf{v})]\right| \\
&\leq \mathbb{E}_{\mathbf{v} \sim \mathbb{B}_1^{A_0}(\mathbf{0})}[|F(\mathbf{x} + \delta\mathbf{v}) - F(\mathbf{y} + \delta\mathbf{v})|] \\
&\leq \mathbb{E}_{\mathbf{v} \sim \mathbb{B}_1^{A_0}(\mathbf{0})}[G\|(\mathbf{x} + \delta\mathbf{v}) - (\mathbf{y} + \delta\mathbf{v})\|] \\
&= G\|\mathbf{x} - \mathbf{y}\|,
\end{aligned}$$

and

$$|\tilde{F}(\mathbf{x}) - F(\mathbf{x})| = |\mathbb{E}_{\mathbf{v}\sim\mathbb{B}_1^{A_0}(\mathbf{0})}[F(\mathbf{x}+\delta\mathbf{v}) - F(\mathbf{x})]|$$
$$\leq \mathbb{E}_{\mathbf{v}\sim\mathbb{B}_1^{A_0}(\mathbf{0})}[|F(\mathbf{x}+\delta\mathbf{v}) - F(\mathbf{x})|]$$
$$\leq \mathbb{E}_{\mathbf{v}\sim\mathbb{B}_1^{A_0}(\mathbf{0})}[G\delta\|\mathbf{v}\|]$$
$$\leq \delta G.$$

If $F$ is $G$-Lipschitz continuous and continuous DR-submodular, then $F$ is differentiable and we have $\nabla F(\mathbf{x}) \geq \nabla F(\mathbf{y})$ for $\forall \mathbf{x} \leq \mathbf{y}$. By definition of $\tilde{F}$, we see that $\tilde{F}$ is also differentiable and

$$\nabla\tilde{F}(\mathbf{x}) - \nabla\tilde{F}(\mathbf{y}) = \nabla\mathbb{E}_{\mathbf{v}\sim\mathbb{B}_1^{A_0}(\mathbf{0})}[F(\mathbf{x}+\delta\mathbf{v})] - \nabla\mathbb{E}_{\mathbf{v}\sim\mathbb{B}_1^{A_0}(\mathbf{0})}[F(\mathbf{y}+\delta\mathbf{v})]$$
$$= \mathbb{E}_{\mathbf{v}\sim\mathbb{B}_1^{A_0}(\mathbf{0})}[\nabla F(\mathbf{x}+\delta\mathbf{v}) - \nabla F(\mathbf{y}+\delta\mathbf{v})]$$
$$\geq \mathbb{E}_{\mathbf{v}\sim\mathbb{B}_1^{A_0}(\mathbf{0})}[0] = 0,$$

for all $\mathbf{x} \leq \mathbf{y}$.

If $F$ is $L$-smooth, then we have $\|\nabla F(\mathbf{x}) - \nabla F(\mathbf{y})\| \leq L\|\mathbf{x}-\mathbf{y}\|$, for all $\mathbf{x},\mathbf{y} \in \mathcal{D}$. Therefore, we have

$$\|\nabla\tilde{F}(\mathbf{x}) - \nabla\tilde{F}(\mathbf{y})\| = \|\nabla\mathbb{E}_{\mathbf{v}\sim\mathbb{B}_1^{A_0}(\mathbf{0})}[F(\mathbf{x}+\delta\mathbf{v})] - \nabla\mathbb{E}_{\mathbf{v}\sim\mathbb{B}_1^{A_0}(\mathbf{0})}[F(\mathbf{y}+\delta\mathbf{v})]\|$$
$$= \|\mathbb{E}_{\mathbf{v}\sim\mathbb{B}_1^{A_0}(\mathbf{0})}[\nabla F(\mathbf{x}+\delta\mathbf{v})] - \mathbb{E}_{\mathbf{v}\sim\mathbb{B}_1^{A_0}(\mathbf{0})}[\nabla F(\mathbf{y}+\delta\mathbf{v})]\|$$
$$\leq \mathbb{E}_{\mathbf{v}\sim\mathbb{B}_1^{A_0}(\mathbf{0})}[\|\nabla F(\mathbf{x}+\delta\mathbf{v}) - \nabla F(\mathbf{y}+\delta\mathbf{v})\|]$$
$$\leq \mathbb{E}_{\mathbf{v}\sim\mathbb{B}_1^{A_0}(\mathbf{0})}[L\|\mathbf{x}-\mathbf{y}\|] = L\|\mathbf{x}-\mathbf{y}\|,$$

for all $\mathbf{x} \leq \mathbf{y}$.

If $F$ is monotone, then we have $F(\mathbf{x}) \leq F(\mathbf{y})$ for all $\mathbf{x} \leq \mathbf{y}$. Therefore

$$\tilde{F}(\mathbf{x}) - \tilde{F}(\mathbf{y}) = \mathbb{E}_{\mathbf{v}\sim\mathbb{B}_1^{A_0}(\mathbf{0})}[F(\mathbf{x}+\delta\mathbf{v})] - \mathbb{E}_{\mathbf{v}\sim\mathbb{B}_1^{A_0}(\mathbf{0})}[F(\mathbf{y}+\delta\mathbf{v})]$$
$$= \mathbb{E}_{\mathbf{v}\sim\mathbb{B}_1^{A_0}(\mathbf{0})}[F(\mathbf{x}+\delta\mathbf{v}) - F(\mathbf{y}+\delta\mathbf{v})]$$
$$\leq \mathbb{E}_{\mathbf{v}\sim\mathbb{B}_1^{A_0}(\mathbf{0})}[0] = 0,$$

for all $\mathbf{x} \leq \mathbf{y}$. Hence $\tilde{F}$ is also monotone. $\qquad\square$

**Lemma 4** (Lemma 10 of [27]). *Let $\mathcal{D} \subseteq \mathbb{R}^d$ such that $\mathrm{aff}(\mathcal{D}) = \mathbb{R}^d$. Assume $F : \mathcal{D} \to \mathbb{R}$ is a $G$-Lipschitz continuous function and let $\tilde{F}$ be its $\delta$-smoothed version. For any $\mathbf{z} \in \mathcal{D}$ such that $\mathbb{B}_\delta(\mathbf{z}) \subseteq \mathcal{D}$, we have*

$$\mathbb{E}_{\mathbf{u}\sim S^{d-1}}\left[\frac{d}{2\delta}(F(\mathbf{z}+\delta\mathbf{u}) - F(\mathbf{z}-\delta\mathbf{u}))\mathbf{u}\right] = \nabla\tilde{F}(\mathbf{z}),$$

$$\mathbb{E}_{\mathbf{u}\sim S^{d-1}}\left[\|\frac{d}{2\delta}(F(\mathbf{z}+\delta\mathbf{u}) - F(\mathbf{z}-\delta\mathbf{u}))\mathbf{u} - \nabla\tilde{F}(\mathbf{z})\|^2\right] \leq CdG^2,$$

*where $C$ is a constant.*

When the convex feasible region $\mathcal{K}$ lies in an affine subspace, we cannot employ the standard spherical sampling method. We extend Lemma 4 to that case.

**Lemma 5.** *Let $\mathcal{D} \subseteq \mathbb{R}^d$ and $A := \mathrm{aff}(\mathcal{D})$. Also let $A_0$ be the translation of $A$ that contains $0$ and let $k = \dim(A)$. Assume $F : \mathcal{D} \to \mathbb{R}$ is a $G$-Lipschitz continuous function and let $\tilde{F}$ be its $\delta$-smoothed version. For any $\mathbf{z} \in \mathcal{D}$ such that $\mathbb{B}_\delta^A(\mathbf{z}) \subseteq \mathcal{D}$, we have*

$$\mathbb{E}_{\mathbf{u}\sim S^{d-1}\cap A_0}\left[\frac{k}{2\delta}(F(\mathbf{z}+\delta\mathbf{u}) - F(\mathbf{z}-\delta\mathbf{u}))\mathbf{u}\right] = \nabla\tilde{F}(\mathbf{z}),$$

$$\mathbb{E}_{\mathbf{u}\sim S^{d-1}\cap A_0}\left[\|\frac{k}{2\delta}(F(\mathbf{z}+\delta\mathbf{u}) - F(\mathbf{z}-\delta\mathbf{u}))\mathbf{u} - \nabla\tilde{F}(\mathbf{z})\|^2\right] \leq CkG^2,$$

*where $C$ is the constant in Lemma 4.*

*Proof.* First consider the case where $A = \mathbb{R}^k \times (0, \cdots, 0)$. In this case, we restrict ourselves to first $k$ coordinates and see that the problem reduces to Lemma 4.

For the general case, let $O$ be an orthonormal transformation that maps $\mathbb{R}^k \times (0, \cdots, 0)$ into $A_0$. Now define $\mathcal{D}' = O^{-1}(\mathcal{D} - \mathbf{z})$ and $F' : \mathcal{D}' \to \mathbb{R} : x \mapsto F(O(x) + \mathbf{z})$. Let $\tilde{F}'$ be the $\delta$-smoothed version of $F'$. Note that $O\left(\nabla \tilde{F}'(0)\right) = \nabla \tilde{F}(\mathbf{z})$. On the other hand, we have

$$\mathrm{aff}(\mathcal{D}') = O^{-1}(A - \mathbf{z}) = O^{-1}(A_0) = \mathbb{R}^k \times (0, \cdots, 0).$$

Therefore

$$\mathbb{E}_{\mathbf{u} \sim S^{d-1} \cap (\mathbb{R}^k \times (0, \cdots, 0))} \left[ \frac{k}{2\delta}(F'(\delta \mathbf{u}) - F'(-\delta \mathbf{u}))\mathbf{u} \right] = \nabla \tilde{F}'(0),$$

and

$$\mathbb{E}_{\mathbf{u} \sim S^{d-1} \cap (\mathbb{R}^k \times (0, \cdots, 0))} \left[ \|\frac{k}{2\delta}(F'(\delta \mathbf{u}) - F'(-\delta \mathbf{u}))\mathbf{u} - \nabla \tilde{F}'(0)\|^2 \right] \leq CkG^2.$$

Hence, if we set $\mathbf{v} = O^{-1}(\mathbf{u})$, we have

$$\mathbb{E}_{\mathbf{u} \sim S^{d-1} \cap A_0} \left[ \frac{k}{2\delta}(F(\mathbf{z} + \delta \mathbf{u}) - F(\mathbf{z} - \delta \mathbf{u}))\mathbf{u} \right]$$

$$= \mathbb{E}_{\mathbf{v} \sim S^{d-1} \cap (\mathbb{R}^k \times (0, \cdots, 0))} \left[ \frac{k}{2\delta}(F'(\delta \mathbf{v}) - F'(-\delta \mathbf{v}))O(\mathbf{v}) \right]$$

$$= O\left( \mathbb{E}_{\mathbf{v} \sim S^{d-1} \cap (\mathbb{R}^k \times (0, \cdots, 0))} \left[ \frac{k}{2\delta}(F'(\delta \mathbf{v}) - F'(-\delta \mathbf{v}))\mathbf{v} \right] \right)$$

$$= O\left( \nabla \tilde{F}'(0) \right)$$

$$= \nabla \tilde{F}(\mathbf{z}).$$

Similarly

$$\mathbb{E}_{\mathbf{u} \sim S^{d-1} \cap A_0} \left[ \|\frac{k}{2\delta}(F(\mathbf{z} + \delta \mathbf{u}) - F(\mathbf{z} - \delta \mathbf{u}))\mathbf{u} - \nabla \tilde{F}(\mathbf{z})\|^2 \right]$$

$$= \mathbb{E}_{\mathbf{v} \sim S^{d-1} \cap (\mathbb{R}^k \times (0, \cdots, 0))} \left[ \|\frac{k}{2\delta}(F'(\delta \mathbf{v}) - F'(-\delta \mathbf{v}))O(\mathbf{v}) - O\left( \nabla \tilde{F}'(0) \right) \|^2 \right]$$

$$= \mathbb{E}_{\mathbf{v} \sim S^{d-1} \cap (\mathbb{R}^k \times (0, \cdots, 0))} \left[ \|O\left( \frac{k}{2\delta}(F'(\delta \mathbf{v}) - F'(-\delta \mathbf{v}))\mathbf{v} - \nabla \tilde{F}'(0) \right) \|^2 \right]$$

$$= \mathbb{E}_{\mathbf{v} \sim S^{d-1} \cap (\mathbb{R}^k \times (0, \cdots, 0))} \left[ \|\frac{k}{2\delta}(F'(\delta \mathbf{v}) - F'(-\delta \mathbf{v}))\mathbf{v} - \nabla \tilde{F}'(0)\|^2 \right]$$

$$\leq CkG^2. \qquad \square$$

*Remark* 4. Note that the same argument may be applied to obtain the one-point gradient estimator:

$$\mathbb{E}_{\mathbf{u} \sim S^{d-1} \cap A_0} \left[ \frac{k}{\delta}F(\mathbf{z} + \delta \mathbf{u})\mathbf{u} \right] = \nabla \tilde{F}(\mathbf{z}).$$

## G   Construction of $\mathcal{K}_\delta$

**Lemma 6.** *Let $\mathcal{K} \subseteq [0, 1]^d$ be a convex set containing the origin. Then for any choice of $\mathbf{c}$ and $r$ with $\mathbb{B}_r^{\mathrm{aff}(\mathcal{K})}(\mathbf{c}) \subseteq \mathcal{K}$, we have*

$$\operatorname*{argmin}_{\mathbf{z} \in \mathcal{K}_\delta} \|\mathbf{z}\|_\infty = \frac{\delta}{r}\mathbf{c} \quad \text{and} \quad \min_{\mathbf{z} \in \mathcal{K}_\delta} \|\mathbf{z}\|_\infty \leq \frac{\delta}{r}.$$

*Proof.* The claim follows immediately from the definition and the fact that $\|\mathbf{c}\|_\infty \leq 1$. $\qquad \square$

**Lemma 7.** *Let $\mathcal{K}$ be an arbitrary convex set, $D := \mathrm{Diam}(\mathcal{K})$ and $\delta' := \frac{\delta D}{r}$. We have*

$$\mathbb{B}_\delta^{\mathrm{aff}(\mathcal{K})}(\mathcal{K}_\delta) \subseteq \mathcal{K} \subseteq \mathbb{B}_{\delta'}^{\mathrm{aff}(\mathcal{K})}(\mathcal{K}_\delta).$$

*Proof.* Define $\psi : \mathcal{K} \to \mathcal{K}_\delta := \mathbf{x} \mapsto (1 - \frac{\delta}{r})\mathbf{x} + \frac{\delta}{r}\mathbf{c}$. Let $\mathbf{y} \in \mathcal{K}_\delta$ and $\mathbf{x} = \psi^{-1}(\mathbf{y})$. Then

$$\mathbb{B}_\delta^{\mathrm{aff}(\mathcal{K})}(\mathbf{y}) = \mathbb{B}_\delta^{\mathrm{aff}(\mathcal{K})}(\psi(\mathbf{x})) = \mathbb{B}_\delta^{\mathrm{aff}(\mathcal{K})}((1 - \frac{\delta}{r})\mathbf{x} + \frac{\delta}{r}\mathbf{c})$$

$$= (1 - \frac{\delta}{r})\mathbf{x} + \mathbb{B}_\delta^{\mathrm{aff}(\mathcal{K})}(\frac{\delta}{r}\mathbf{c}) = (1 - \frac{\delta}{r})\mathbf{x} + \frac{\delta}{r}\mathbb{B}_r^{\mathrm{aff}(\mathcal{K})}(\mathbf{c}) \subseteq \mathcal{K},$$

where the last inclusion follows from the fact that $\mathcal{K}$ is convex and contains both $\mathbf{x}$ and $\mathbb{B}_r^{\mathrm{aff}(\mathcal{K})}(\mathbf{c})$. On the other hand, for any $\mathbf{x} \in \mathcal{K} \subseteq \mathrm{aff}(\mathcal{K})$, we have

$$\|\psi(\mathbf{x}) - \mathbf{x}\| = \frac{\delta}{r}\|\mathbf{x} - \mathbf{c}\| < \frac{\delta}{r}D = \delta'.$$

Therefore

$$\mathbf{x} \in \mathbb{B}_{\delta'}(\psi(\mathbf{x})) \cap \mathrm{aff}(\mathcal{K}) = \mathbb{B}_{\delta'}^{\mathrm{aff}(\mathcal{K})}(\psi(\mathbf{x})) \subseteq \mathbb{B}_{\delta'}^{\mathrm{aff}(\mathcal{K})}(\mathcal{K}_\delta). \qquad \square$$

**Choice of c and $r$**    While the results hold for any choice of $\mathbf{c} \in \mathcal{K}$ and $r$ with $\mathbb{B}_r^{\mathrm{aff}}(\mathbf{c}) \subseteq \mathcal{K}$, as can be seen in Theorem 2, the approximation errors depends linearly on $1/r$. Therefore, it is natural to choose the point $\mathbf{c}$ that maximizes the value of $r$, the *Chebyshev center* of $\mathcal{K}$.

**Analytic Constraint Model — Polytope**    When the feasible region $\mathcal{K}$ is characterized by a set of $q$ linear constraints $\mathbf{A}\mathbf{x} \le \mathbf{b}$ with a known coefficient matrix $\mathbf{A} \in \mathbb{R}^{q \times d}$ and vector $\mathbf{b} \in \mathbb{R}^q$, thus $\mathcal{K}$ is a polytope, by the linearity of the transformation (7), the shrunken feasible region $\mathcal{K}_\delta$ is similarly characterized by a (translated) set of $q$ linear constraints $\mathbf{A}\mathbf{x} \le (1 - \frac{\delta}{r})\mathbf{b} + \frac{\delta}{r}\mathbf{A}\mathbf{c}$.

## H    Variance reduction via momentum

In order to prove main regret bounds, we need the following variance reduction lemma, which is crucial in characterizing how much the variance of the gradient estimator can be reduced by using momentum. This lemma appears in [6] and it is a slight improvement of Lemma 2 in [22] and Lemma 5 in [23].

**Lemma 8** (Theorem 3 of [6]). *Let $\{\mathbf{a}_n\}_{n=0}^N$ be a sequence of points in $\mathbb{R}^d$ such that $\|\mathbf{a}_n - \mathbf{a}_{n-1}\| \le G_0/(n+s)$ for all $1 \le n \le N$ with fixed constants $G_0 \ge 0$ and $s \ge 3$. Let $\{\tilde{\mathbf{a}}_n\}_{n=1}^N$ be a sequence of random variables such that $\mathbb{E}[\tilde{\mathbf{a}}_n | \mathcal{F}_{n-1}] = \mathbf{a}_n$ and $\mathbb{E}[\|\tilde{\mathbf{a}}_n - \mathbf{a}_n\|^2 | \mathcal{F}_{n-1}] \le \sigma^2$ for every $n \ge 0$, where $\mathcal{F}_{n-1}$ is the $\sigma$-field generated by $\{\tilde{\mathbf{a}}_i\}_{i=1}^n$ and $\mathcal{F}_0 = \varnothing$. Let $\{\mathbf{d}_n\}_{n=0}^N$ be a sequence of random variables where $\mathbf{d}_0$ is fixed and subsequent $\mathbf{d}_n$ are obtained by the recurrence*

$$\mathbf{d}_n = (1 - \rho_n)\mathbf{d}_{n-1} + \rho_n \tilde{\mathbf{a}}_n \tag{17}$$

*with $\rho_n = \frac{2}{(n+s)^{2/3}}$. Then, we have*

$$\mathbb{E}[\|\mathbf{a}_n - \mathbf{d}_n\|^2] \le \frac{Q}{(n+s+1)^{2/3}}, \tag{18}$$

*where $Q := \max\{\|\mathbf{a}_0 - \mathbf{d}_0\|^2(s+1)^{2/3}, 4\sigma^2 + 3G_0^2/2\}$.*

We now analyze the variance of our gradient estimator, which, in the case when we only have access zeroth-order information, uses batched spherical sampling and momentum for gradient estimation. Calculations similar to the proof of the following Lemma, in the value oracle case, appear in the proof of Theorem 2 in [8]. The main difference is that here we consider a more general smoothing trick and therefore we estimate the gradient along the affine hull of $\mathcal{K}$.

**Lemma 9.** *Under the assumptions of Theorem 1, in Algorithm 2, we have*

$$\mathbb{E}\left[\|\nabla(\tilde{F}|_{\mathcal{L}})(\mathbf{z}_n) - \bar{\mathbf{g}}_n\|^2\right] \le \frac{Q}{(n+4)^{2/3}},$$

*for all* $1 \leq n \leq N$ *where* $\mathcal{L} = \mathrm{aff}(\mathcal{K})$,

$$Q = \begin{cases} 0 & \textit{det. grad. oracle,} \\ \max\{4^{2/3}G^2, 6L^2D^2 + \frac{4\sigma_1^2}{B}\} & \textit{stoch. grad. oracle with variance } \sigma_1^2 > 0, \\ \max\{4^{2/3}G^2, 6L^2D^2 + \frac{4CkG^2 + 2k^2\sigma_0^2/\delta^2}{B}\} & \textit{value oracle with variance } \sigma_0^2 \geq 0, \end{cases}$$

$C$ *is a constant and* $D = \mathrm{diam}(\mathcal{K})$.

*Remark* 5. As we will see in the proof of Theorem 1, except for the case with deterministic gradient oracle, the dominating term in the approximation error is a constant multiple of

$$\frac{1}{N}\sum_{n=1}^{N} \mathbb{E}\left[\|\nabla(\tilde{F}|_{\mathcal{L}})(\mathbf{z}_n) - \bar{\mathbf{g}}_n\|^2\right].$$

Therefore, any improvement in Lemma 9 will result in direct improvement of the approximation error.

*Proof.* If we have access to a deterministic gradient oracle, then the claim is trivial. Let $\mathcal{F}_1 := \varnothing$ and $\mathcal{F}_n$ be the $\sigma$-field generated by $\{\bar{\mathbf{g}}_1, \ldots, \bar{\mathbf{g}}_{n-1}\}$ and let

$$\sigma^2 = \begin{cases} \frac{\sigma_1^2}{B} & \text{stoch. grad. oracle with variance } \sigma_1^2 > 0, \\ \frac{CkG^2 + k^2\sigma_0^2/2\delta^2}{B} & \text{value oracle with variance } \sigma_0^2 \geq 0. \end{cases}$$

Let $\mathcal{L}_0$ denote the linear space $\mathcal{L} - \mathbf{x}$ for some $\mathbf{x} \in \mathcal{L}$. If we have access to a stochastic gradient oracle, then $\mathbf{g}_n$ is computed by taking the average of $B$ gradient samples of $P_{\mathcal{L}_0}(\hat{G}(\mathbf{z}))$, i.e. the projection of $\hat{G}(\mathbf{z})$ onto the linear space $\mathcal{L}_0$. Since $P_{\mathcal{L}_0}$ is a 1-Lipscitz linear map, we see that

$$\mathbb{E}[P_{\mathcal{L}_0}(\hat{G}(\mathbf{z}))] = P_{\mathcal{L}_0}(\nabla\tilde{F}(\mathbf{z})) = \nabla(\tilde{F}|_{\mathcal{L}})(\mathbf{z})$$

and

$$\mathbb{E}\left[\left\|P_{\mathcal{L}_0}(\hat{G}(\mathbf{z})) - \nabla(\tilde{F}|_{\mathcal{L}})(\mathbf{z})\right\|^2\right] = \mathbb{E}\left[\left\|P_{\mathcal{L}_0}(\hat{G}(\mathbf{z})) - P_{\mathcal{L}_0}(\nabla\tilde{F}(\mathbf{z}))\right\|^2\right]$$

$$\leq \mathbb{E}\left[\left\|\hat{G}(\mathbf{z}) - \nabla\tilde{F}(\mathbf{z})\right\|^2\right] \leq \sigma_1^2.$$

Note that, in cases where we have access to a gradient oracle, we have $\delta = 0$ and $\tilde{F} = F$. Therefore

$$\mathbb{E}\left[\mathbf{g}_n|\mathcal{F}_{n-1}\right] = \nabla(\tilde{F}|_{\mathcal{L}})(\mathbf{z}_n) \quad \text{and} \quad \mathbb{E}\left[\|\mathbf{g}_n - \nabla(\tilde{F}|_{\mathcal{L}})(\mathbf{z}_n)\|^2|\mathcal{F}_{n-1}\right] \leq \frac{\sigma_1^2}{B} = \sigma^2.$$

Next we assume that we have access to a value oracle. By the unbiasedness of $\hat{F}$ and Lemma 5, we have

$$\mathbb{E}\left[\frac{k}{2\delta}(\hat{F}(\mathbf{y}_{n,i}^+) - \hat{F}(\mathbf{y}_{n,i}^-))\mathbf{u}_{n,i}|\mathcal{F}_{n-1}\right] = \mathbb{E}\left[\mathbb{E}\left[\frac{k}{2\delta}(\hat{F}(\mathbf{y}_{n,i}^+) - \hat{F}(\mathbf{y}_{n,i}^-))\mathbf{u}_{n,i}|\mathcal{F}_{n-1}, \mathbf{u}_{n,i}\right]|\mathcal{F}_{n-1}\right]$$

$$= \mathbb{E}\left[\frac{k}{2\delta}(F(\mathbf{y}_{n,i}^+) - F(\mathbf{y}_{n,i}^-))\mathbf{u}_{n,i}|\mathcal{F}_{n-1}\right]$$

$$= \nabla(\tilde{F}|_{\mathcal{L}})(\mathbf{z}_n),$$

and

$$\mathbb{E}\left[\left\|\frac{k}{2\delta}(\hat{F}(\mathbf{y}_{n,i}^+) - \hat{F}(\mathbf{y}_{n,i}^-))\mathbf{u}_{n,i} - \nabla(\tilde{F}|_{\mathcal{L}})(\mathbf{z}_n)\right\|^2 |\mathcal{F}_{n-1}\right]$$

$$= \mathbb{E}\left[\mathbb{E}\left[\left\|\frac{k}{2\delta}(F(\mathbf{y}_{n,i}^+) - F(\mathbf{y}_{n,i}^-))\mathbf{u}_{n,i} - \nabla(\tilde{F}|_{\mathcal{L}})(\mathbf{z}_n)\right.\right.\right.$$

$$+ \frac{k}{2\delta}(\hat{F}(\mathbf{y}_{n,i}^+) - F(\mathbf{y}_{n,i}^+))\mathbf{u}_{n,i}$$

$$\left.\left.\left.- \frac{k}{2\delta}(\hat{F}(\mathbf{y}_{n,i}^-) - F(\mathbf{y}_{n,i}^-))\mathbf{u}_{n,i}\right\|^2 |\mathcal{F}_{n-1}, \mathbf{u}_{n,i}\right]|\mathcal{F}_{n-1}\right]$$

$$\leq \mathbb{E}\left[\mathbb{E}\left[\left\|\frac{k}{2\delta}(F(\mathbf{y}_{n,i}^+) - F(\mathbf{y}_{n,i}^-))\mathbf{u}_{n,i} - \nabla(\tilde{F}|_{\mathcal{L}})(\mathbf{z}_n)\right\|^2 |\mathcal{F}_{n-1}, \mathbf{u}_{n,i}\right]|\mathcal{F}_{n-1}\right]$$

$$+ \mathbb{E}\left[\mathbb{E}\left[\left\|\frac{k}{2\delta}(\hat{F}(\mathbf{y}_{n,i}^+) - F(\mathbf{y}_{n,i}^+))\mathbf{u}_{n,i}\right\|^2 |\mathcal{F}_{n-1}, \mathbf{u}_{n,i}\right]|\mathcal{F}_{n-1}\right]$$

$$+ \mathbb{E}\left[\mathbb{E}\left[\left\|\frac{k}{2\delta}(\hat{F}(\mathbf{y}_{n,i}^-) - F(\mathbf{y}_{n,i}^-))\mathbf{u}_{n,i}\right\|^2 |\mathcal{F}_{n-1}, \mathbf{u}_{n,i}\right]|\mathcal{F}_{n-1}\right]$$

$$\leq \mathbb{E}\left[\left\|\frac{k}{2\delta}(F(\mathbf{y}_{n,i}^+) - F(\mathbf{y}_{n,i}^-))\mathbf{u}_{n,i} - \nabla(\tilde{F}|_{\mathcal{L}})(\mathbf{z}_n)\right\|^2 |\mathcal{F}_{n-1}\right]$$

$$+ \frac{k^2}{4\delta^2}\mathbb{E}\left[\mathbb{E}\left[|\hat{F}(\mathbf{y}_{n,i}^+) - F(\mathbf{y}_{n,i}^+)|^2 \cdot \|\mathbf{u}_{n,i}\|^2 |\mathcal{F}_{n-1}, \mathbf{u}_{n,i}\right]|\mathcal{F}_{n-1}\right]$$

$$+ \frac{k^2}{4\delta^2}\mathbb{E}\left[\mathbb{E}\left[|\hat{F}(\mathbf{y}_{n,i}^-) - F(\mathbf{y}_{n,i}^-)|^2 \cdot \|\mathbf{u}_{n,i}\|^2 |\mathcal{F}_{n-1}, \mathbf{u}_{n,i}\right]|\mathcal{F}_{n-1}\right]$$

$$\leq CkG^2 + \frac{k^2}{4\delta^2}\sigma_0^2 + \frac{k^2}{4\delta^2}\sigma_0^2$$

$$= CkG^2 + \frac{k^2}{2\delta^2}\sigma_0^2.$$

So we have

$$\mathbb{E}\left[\mathbf{g}_n|\mathcal{F}_{n-1}\right] = \mathbb{E}\left[\frac{1}{B}\sum_{i=1}^{B}\frac{k}{2\delta}(\hat{F}(\mathbf{y}_{n,i}^+) - \hat{F}(\mathbf{y}_{n,i}^-))\mathbf{u}_{n,i}|\mathcal{F}_{n-1}\right] = \nabla(\tilde{F}|_{\mathcal{L}})(\mathbf{z}_n),$$

and

$$\mathbb{E}\left[\left\|\mathbf{g}_n - \nabla(\tilde{F}|_{\mathcal{L}})(\mathbf{z}_n)\right\|^2 |\mathcal{F}_{n-1}\right]$$

$$= \frac{1}{B^2}\sum_{i=1}^{B}\mathbb{E}\left[\left\|\frac{k}{2\delta}(\hat{F}(\mathbf{y}_{n,i}^+) - \hat{F}(\mathbf{y}_{n,i}^-))\mathbf{u}_{n,i} - \nabla(\tilde{F}|_{\mathcal{L}})(\mathbf{z}_n)\right\|^2 |\mathcal{F}_{n-1}\right]$$

$$\leq \frac{CkG^2 + \frac{k^2}{2\delta^2}\sigma_0^2}{B} = \sigma^2.$$

Using Lemma 8 with $\mathbf{d}_n = \bar{\mathbf{g}}_n, \tilde{\mathbf{a}}_n = \mathbf{g}_n, \mathbf{a}_n = \nabla(\tilde{F}|_{\mathcal{L}})(\mathbf{z}_n)$ for all $n \geq 1$, $\mathbf{a}_0 = \nabla(\tilde{F}|_{\mathcal{L}})(\mathbf{z}_1)$, $G_0 = 2LD$ and $s = 3$, we have

$$\mathbb{E}[\|\nabla(\tilde{F}|_{\mathcal{L}})(\mathbf{z}_n) - \bar{\mathbf{g}}_n\|^2] \leq \frac{Q'}{(n+4)^{2/3}}, \tag{19}$$

where $Q' = \max\{\|\nabla(\tilde{F}|_{\mathcal{L}})(\mathbf{z}_1)\|^2 4^{2/3}, 6L^2D^2 + 4\sigma^2\}$. Note that by Lemma 3, we have $\|\nabla(\tilde{F}|_{\mathcal{L}})(x)\| \leq G$, thus we have $Q' \leq Q$. $\qquad\square$

# I  Proof of Theorem 1 for monotone maps over convex sets containing zero

*Proof.* By the definition of $\mathbf{z}_n$, we have $\mathbf{z}_n = \mathbf{z}_1 + \sum_{i=1}^{n-1} \frac{\mathbf{v}_i}{N}$. Therefore $\mathbf{z}_n - \mathbf{z}_1$ is a convex combination of $\mathbf{v}_n$'s and 0 which belong to $\mathcal{K}_\delta - \mathbf{z}_1$ and therefore $\mathbf{z}_n - \mathbf{z}_1 \in \mathcal{K}_\delta - \mathbf{z}_1$. Hence we have $\mathbf{z}_n \in \mathcal{K}_\delta \subseteq \mathcal{K}$ for all $1 \leq n \leq N+1$.

Let $\mathcal{L} := \text{aff}(\mathcal{K})$. According to Lemma 3, the function $\tilde{F}$ is $L$-smooth. So we have

$$
\begin{aligned}
\tilde{F}(\mathbf{z}_{n+1}) - \tilde{F}(\mathbf{z}_n) &\geq \langle \nabla(\tilde{F}|_{\mathcal{L}})(\mathbf{z}_n), \mathbf{z}_{n+1} - \mathbf{z}_n \rangle - \frac{L}{2}\|\mathbf{z}_{n+1} - \mathbf{z}_n\|^2 \\
&= \varepsilon \langle \nabla(\tilde{F}|_{\mathcal{L}})(\mathbf{z}_n), \mathbf{v}_n \rangle - \frac{\varepsilon^2 L}{2}\|\mathbf{v}_n\|^2 \\
&\geq \varepsilon \langle \nabla(\tilde{F}|_{\mathcal{L}})(\mathbf{z}_n), \mathbf{v}_n \rangle - \frac{\varepsilon^2 L}{2}D^2 \\
&= \varepsilon \left( \langle \bar{\mathbf{g}}_n, \mathbf{v}_n \rangle + \langle \nabla(\tilde{F}|_{\mathcal{L}})(\mathbf{z}_n) - \bar{\mathbf{g}}_n, \mathbf{v}_n \rangle \right) - \frac{\varepsilon^2 L D^2}{2}.
\end{aligned}
\tag{20}
$$

Let $\mathbf{z}_\delta^* := \text{argmax}_{\mathbf{z} \in \mathcal{K}_\delta - \mathbf{z}_1} \tilde{F}(z)$. We have $\mathbf{z}_\delta^* \in \mathcal{K}_\delta - \mathbf{z}_1$, which implies that $\langle \bar{\mathbf{g}}_n, \mathbf{v}_n \rangle \geq \langle \bar{\mathbf{g}}_n, \mathbf{z}_\delta^* \rangle$. Therefore

$$
\langle \bar{\mathbf{g}}_n, \mathbf{v}_n \rangle \geq \langle \bar{\mathbf{g}}_n, \mathbf{z}_\delta^* \rangle = \langle \nabla(\tilde{F}|_{\mathcal{L}})(\mathbf{z}_n), \mathbf{z}_\delta^* \rangle + \langle \bar{\mathbf{g}}_n - \nabla(\tilde{F}|_{\mathcal{L}})(\mathbf{z}_n), \mathbf{z}_\delta^* \rangle
$$

Hence we obtain

$$
\langle \bar{\mathbf{g}}_n, \mathbf{v}_n \rangle + \langle \nabla(\tilde{F}|_{\mathcal{L}})(\mathbf{z}_n) - \bar{\mathbf{g}}_n, \mathbf{v}_n \rangle \geq \langle \nabla(\tilde{F}|_{\mathcal{L}})(\mathbf{z}_n), \mathbf{z}_\delta^* \rangle - \langle \nabla(\tilde{F}|_{\mathcal{L}})(\mathbf{z}_n) - \bar{\mathbf{g}}_n, \mathbf{z}_\delta^* - \mathbf{v}_n \rangle
$$

Using the Cauchy-Schwartz inequality, we have

$$
\langle \nabla(\tilde{F}|_{\mathcal{L}})(\mathbf{z}_n) - \bar{\mathbf{g}}_n, \mathbf{z}_\delta^* - \mathbf{v}_n \rangle \leq \|\nabla(\tilde{F}|_{\mathcal{L}})(\mathbf{z}_n) - \bar{\mathbf{g}}_n\|\|\mathbf{z}_\delta^* - \mathbf{v}_n\| \leq D\|\nabla(\tilde{F}|_{\mathcal{L}})(\mathbf{z}_n) - \bar{\mathbf{g}}_n\|
$$

Therefore

$$
\langle \bar{\mathbf{g}}_n, \mathbf{v}_n \rangle + \langle \nabla(\tilde{F}|_{\mathcal{L}})(\mathbf{z}_n) - \bar{\mathbf{g}}_n, \mathbf{v}_n \rangle \geq \langle \nabla(\tilde{F}|_{\mathcal{L}})(\mathbf{z}_n), \mathbf{z}_\delta^* \rangle - D\|\nabla(\tilde{F}|_{\mathcal{L}})(\mathbf{z}_n) - \bar{\mathbf{g}}_n\|.
$$

Plugging this into 20, we see that

$$
\tilde{F}(\mathbf{z}_{n+1}) - \tilde{F}(\mathbf{z}_n) \geq \varepsilon \langle \nabla(\tilde{F}|_{\mathcal{L}})(\mathbf{z}_n), \mathbf{z}_\delta^* \rangle - \varepsilon D\|\nabla(\tilde{F}|_{\mathcal{L}})(\mathbf{z}_n) - \bar{\mathbf{g}}_n\| - \frac{\varepsilon^2 L D^2}{2}.
\tag{21}
$$

On the other hand, we have $\mathbf{z}_\delta^* \geq (\mathbf{z}_\delta^* - \mathbf{z}_n) \vee 0$. Since $F$ is monotone continuous DR-submodular, by Lemma 3, so is $\tilde{F}$. Moreover monotonicity of $\tilde{F}$ implies that $\nabla(\tilde{F}|_{\mathcal{L}})$ is non-negative in positive directions. Therefore we have

$$
\begin{aligned}
\langle \nabla(\tilde{F}|_{\mathcal{L}})(\mathbf{z}_n), \mathbf{z}_\delta^* \rangle &\geq \langle \nabla(\tilde{F}|_{\mathcal{L}})(\mathbf{z}_n), (\mathbf{z}_\delta^* - \mathbf{z}_n) \vee 0 \rangle & \text{(monotonicity)} \\
&\geq \tilde{F}(\mathbf{z}_n + ((\mathbf{z}_\delta^* - \mathbf{z}_n) \vee 0)) - \tilde{F}(\mathbf{z}_n) & \text{(DR-submodularity)} \\
&= \tilde{F}(\mathbf{z}_\delta^* \vee \mathbf{z}_n) - \tilde{F}(\mathbf{z}_n) \\
&\geq \tilde{F}(\mathbf{z}_\delta^*) - \tilde{F}(\mathbf{z}_n)
\end{aligned}
$$

After plugging this into (21) and re-arranging terms, we obtain

$$
h_{n+1} \leq (1-\varepsilon)h_n + \varepsilon D\|\nabla(\tilde{F}|_{\mathcal{L}})(\mathbf{z}_n) - \bar{\mathbf{g}}_n\| + \frac{\varepsilon^2 L D^2}{2}
$$

where $h_n := \tilde{F}(\mathbf{z}_\delta^*) - \tilde{F}(\mathbf{z}_n)$. After taking the expectation and using Lemma 9, we see that

$$
\mathbb{E}(h_{n+1}) \leq (1-\varepsilon)\mathbb{E}(h_n) + \frac{\varepsilon D Q^{1/2}}{(n+4)^{1/3}} + \frac{\varepsilon^2 L D^2}{2}.
$$

Using the above inequality recursively and $1 - \varepsilon \leq 1$, we have

$$
\mathbb{E}[h_{N+1}] \leq (1-\varepsilon)^N \mathbb{E}[h_1] + \sum_{n=1}^{N} \frac{\varepsilon D Q^{1/2}}{(n+4)^{1/3}} + \frac{N\varepsilon^2 L D^2}{2}.
$$

Note that we have $\varepsilon = 1/N$. Using the fact that $(1 - \frac{1}{N})^N \le e^{-1}$ and

$$\sum_{n=1}^{N} \frac{DQ^{1/2}}{(n+4)^{1/3}} \le DQ^{1/2} \int_0^N \frac{\mathrm{d}x}{(x+4)^{1/3}} \le DQ^{1/2}\left(\frac{3}{2}(N+4)^{2/3}\right)$$
$$\le DQ^{1/2}\left(\frac{3}{2}(2N)^{2/3}\right) \le 3DQ^{1/2}N^{2/3}, \tag{22}$$

we see that

$$\mathbb{E}[h_{N+1}] \le e^{-1}\mathbb{E}[h_1] + \frac{3DQ^{1/2}}{N^{1/3}} + \frac{LD^2}{2N}.$$

By re-arranging the terms and using the fact that $\tilde{F}$ is non-negative, we conclude

$$(1 - e^{-1})\tilde{F}(\mathbf{z}_\delta^*) - \mathbb{E}[\tilde{F}(\mathbf{z}_{N+1})] \le -e^{-1}\tilde{F}(\mathbf{z}_1) + \frac{3DQ^{1/2}}{N^{1/3}} + \frac{LD^2}{2N}$$
$$\le \frac{3DQ^{1/2}}{N^{1/3}} + \frac{LD^2}{2N}. \tag{23}$$

According to Lemma 3, we have $\tilde{F}(\mathbf{z}_{N+1}) \le F(\mathbf{z}_{N+1}) + \delta G$. Moreover, using Lemma 7, we see that $\mathbf{z}^* \in \mathbb{B}_{\delta'}(\mathcal{K}_\delta)$ where $\delta' = \delta D/r$. Therefore, there is a point $\mathbf{y}^* \in \mathcal{K}_\delta$ such that $\|\mathbf{y}^* - \mathbf{z}^*\| \le \delta'$.

$$\tilde{F}(\mathbf{z}_\delta^*) \ge \tilde{F}(\mathbf{y}^* - \mathbf{z}_1) \ge \tilde{F}(\mathbf{y}^*) - G\|\mathbf{z}_1\|$$
$$\ge F(\mathbf{y}^*) - (\|\mathbf{z}_1\| + \delta)G \ge F(\mathbf{z}^*) - (\|\mathbf{z}_1\| + \delta + \frac{\delta D}{r})G.$$

According to Lemma 6, we have $\|\mathbf{z}_1\| \le \sqrt{d}\|\mathbf{z}_1\|_\infty \le \delta\sqrt{d}/r$.

$$\tilde{F}(\mathbf{z}_\delta^*) \ge F(\mathbf{z}^*) - (1 + \frac{\sqrt{d} + D}{r})\delta G. \tag{24}$$

After plugging these into 23, we see that

$$(1 - e^{-1})F(\mathbf{z}^*) - \mathbb{E}[F(\mathbf{z}_{N+1})]$$
$$\le \frac{3DQ^{1/2}}{N^{1/3}} + \frac{LD^2}{2N} + \delta G(2 + \frac{\sqrt{d} + D}{r}). \qquad \square$$

## J Proof of Theorem 1 for non-monotone maps over downward-closed convex sets

*Proof.* Similar to Appendix I, we see that $\mathbf{z}_n \in \mathcal{K}_\delta$ for all $1 \le n \le N+1$ and

$$\tilde{F}(\mathbf{z}_{n+1}) - \tilde{F}(\mathbf{z}_n) \ge \varepsilon\left(\langle\bar{\mathbf{g}}_n, \mathbf{v}_n\rangle + \langle\nabla(\tilde{F}|_\mathcal{L})(\mathbf{z}_n) - \bar{\mathbf{g}}_n, \mathbf{v}_n\rangle\right) - \frac{\varepsilon^2 LD^2}{2}. \tag{25}$$

Let $\mathbf{z}_\delta^* := \operatorname{argmax}_{\mathbf{z} \in \mathcal{K}_\delta - \mathbf{z}_1} \tilde{F}(z)$. We have $\mathbf{z}_\delta^* \vee \mathbf{z}_n - \mathbf{z}_n = (\mathbf{z}_\delta^* - \mathbf{z}_n) \vee 0 \le \mathbf{z}_\delta^*$. Therefore, since $\mathcal{K}_\delta$ is downward-closed, we have $\mathbf{z}_\delta^* \vee \mathbf{z}_n - \mathbf{z}_n \in \mathcal{K}_\delta - \mathbf{z}_1$. On the other hand, $\mathbf{z}_\delta^* \vee \mathbf{z}_n - \mathbf{z}_n \le \mathbf{1} - \mathbf{z}_n$. Therefore, we have $\langle\bar{\mathbf{g}}_n, \mathbf{v}_n\rangle \ge \langle\bar{\mathbf{g}}_n, \mathbf{z}_\delta^* \vee \mathbf{z}_n - \mathbf{z}_n\rangle$, which implies that

$$\langle\bar{\mathbf{g}}_n, \mathbf{z}_\delta^* \vee \mathbf{z}_n - \mathbf{z}_n\rangle + \langle\nabla(\tilde{F}|_\mathcal{L})(\mathbf{z}_n) - \bar{\mathbf{g}}_n, \mathbf{v}_n\rangle$$
$$= \langle\nabla(\tilde{F}|_\mathcal{L})(\mathbf{z}_n), \mathbf{z}_\delta^* \vee \mathbf{z}_n - \mathbf{z}_n\rangle + \langle\bar{\mathbf{g}}_n - \nabla(\tilde{F}|_\mathcal{L})(\mathbf{z}_n), \mathbf{z}_\delta^* \vee \mathbf{z}_n - \mathbf{z}_n\rangle$$
$$+ \langle\nabla(\tilde{F}|_\mathcal{L})(\mathbf{z}_n) - \bar{\mathbf{g}}_n, \mathbf{v}_n\rangle$$
$$= \langle\nabla(\tilde{F}|_\mathcal{L})(\mathbf{z}_n), \mathbf{z}_\delta^* \vee \mathbf{z}_n - \mathbf{z}_n\rangle - \langle\nabla(\tilde{F}|_\mathcal{L})(\mathbf{z}_n) - \bar{\mathbf{g}}_n, -\mathbf{v}_n + \mathbf{z}_\delta^* \vee \mathbf{z}_n - \mathbf{z}_n\rangle$$

Using the Cauchy-Shwarz inequality, we see that

$$\langle\nabla(\tilde{F}|_\mathcal{L})(\mathbf{z}_n) - \bar{\mathbf{g}}_n, -\mathbf{v}_n + \mathbf{z}_\delta^* \vee \mathbf{z}_n - \mathbf{z}_n\rangle \le \|\nabla(\tilde{F}|_\mathcal{L})(\mathbf{z}_n) - \bar{\mathbf{g}}_n\|\|(\mathbf{z}_\delta^* \vee \mathbf{z}_n - \mathbf{z}_n) - \mathbf{v}_n\|$$
$$\le D\|\nabla(\tilde{F}|_\mathcal{L})(\mathbf{z}_n) - \bar{\mathbf{g}}_n\|.$$

where the last inequality follows from the fact that both $\mathbf{v}_n$ and $\mathbf{z}_\delta^* \vee \mathbf{z}_n - \mathbf{z}_n$ belong to $\mathcal{K}_\delta$. Therefore

$$\langle \bar{\mathbf{g}}_n, \mathbf{z}_\delta^* \vee \mathbf{z}_n - \mathbf{z}_n \rangle + \langle \nabla(\tilde{F}|_{\mathcal{L}})(\mathbf{z}_n) - \bar{\mathbf{g}}_n, \mathbf{v}_n \rangle$$
$$\geq \langle \nabla(\tilde{F}|_{\mathcal{L}})(\mathbf{z}_n), \mathbf{z}_\delta^* \vee \mathbf{z}_n - \mathbf{z}_n \rangle - D\|\nabla(\tilde{F}|_{\mathcal{L}})(\mathbf{z}_n) - \bar{\mathbf{g}}_n\|.$$

Plugging this into Equation (25), we get

$$\tilde{F}(\mathbf{z}_{n+1}) - \tilde{F}(\mathbf{z}_n)$$
$$\geq \varepsilon \langle \nabla(\tilde{F}|_{\mathcal{L}})(\mathbf{z}_n), \mathbf{z}_\delta^* \vee \mathbf{z}_n - \mathbf{z}_n \rangle - \varepsilon D\|\nabla(\tilde{F}|_{\mathcal{L}})(\mathbf{z}_n) - \bar{\mathbf{g}}_n\| - \frac{\varepsilon^2 L D^2}{2}. \qquad (26)$$

Next we show that

$$1 - \|\mathbf{z}_n\|_\infty \geq (1-\varepsilon)^{n-1}(1 - \frac{\delta}{r}), \qquad (27)$$

for all $1 \leq n \leq N+1$. We use induction on $n$ to show that for each coordinate $1 \leq i \leq d$, we have $1 - [\mathbf{z}_n]_i \geq (1-\varepsilon)^{n-1}$. For $n = 1$, the claim follows from Lemma 6. Assuming that the inequality is true for $n$, using the fact that $\mathbf{v}_n \leq \mathbf{1} - \mathbf{z}_n$, we have

$$1 - [\mathbf{z}_{n+1}]_i = 1 - [\mathbf{z}_n]_i - \varepsilon[\mathbf{v}_n]_i \geq 1 - [\mathbf{z}_n]_i - \varepsilon(1 - [\mathbf{z}_n]_i)$$
$$= (1-\varepsilon)(1 - [\mathbf{z}_n]_i) \geq (1-\varepsilon)^n(1 - \frac{\delta}{r}),$$

which completes the proof by induction.

Since $\tilde{F}$ is DR-submodular, it is concave along non-negative directions. Therefore, using Lemma 1 and Equation (27), we have

$$\langle \nabla(\tilde{F}|_{\mathcal{L}})(\mathbf{z}_n), \mathbf{z}_\delta^* \vee \mathbf{z}_n - \mathbf{z}_n \rangle \geq \tilde{F}(\mathbf{z}_\delta^* \vee \mathbf{z}_n) - \tilde{F}(\mathbf{z}_n)$$
$$\geq (1 - \|\mathbf{z}_n\|_\infty)\tilde{F}(\mathbf{z}_\delta^*) - \tilde{F}(\mathbf{z}_n)$$
$$\geq (1-\varepsilon)^{n-1}(1 - \frac{\delta}{r})\tilde{F}(\mathbf{z}_\delta^*) - \tilde{F}(\mathbf{z}_n).$$

Plugging this into Equation (26), we get

$$\tilde{F}(\mathbf{z}_{n+1}) - \tilde{F}(\mathbf{z}_n)$$
$$\geq \varepsilon \left( (1-\varepsilon)^{n-1}(1 - \frac{\delta}{r})\tilde{F}(\mathbf{z}_\delta^*) - \tilde{F}(\mathbf{z}_n) \right) - \varepsilon D\|\nabla(\tilde{F}|_{\mathcal{L}})(\mathbf{z}_n) - \bar{\mathbf{g}}_n\| - \frac{\varepsilon^2 L D^2}{2}.$$

Taking expectations of both sides and using Lemma 9, we see that

$$\mathbb{E}(\tilde{F}(\mathbf{z}_{n+1})) \geq (1-\varepsilon)\mathbb{E}(\tilde{F}(\mathbf{z}_n)) + \varepsilon(1-\varepsilon)^{n-1}(1 - \frac{\delta}{r})\tilde{F}(\mathbf{z}_\delta^*) - \frac{\varepsilon D Q^{1/2}}{(n+4)^{1/3}} - \frac{\varepsilon^2 L D^2}{2}.$$

Using this inequality recursively and Equation (22), we get

$$\mathbb{E}(\tilde{F}(\mathbf{z}_{N+1})) \geq (1-\varepsilon)^N \mathbb{E}(\tilde{F}(\mathbf{z}_1)) + N\varepsilon(1-\varepsilon)^{N-1}(1 - \frac{\delta}{r})\tilde{F}(\mathbf{z}_\delta^*)$$
$$- \sum_{n=1}^N \frac{\varepsilon D Q^{1/2}}{(n+4)^{1/3}} - \frac{N\varepsilon^2 L D^2}{2}$$
$$\geq (1-\varepsilon)^N \mathbb{E}(\tilde{F}(\mathbf{z}_1)) + N\varepsilon(1-\varepsilon)^{N-1}(1 - \frac{\delta}{r})\tilde{F}(\mathbf{z}_\delta^*)$$
$$- 3\varepsilon D Q^{1/2} N^{2/3} - \frac{N\varepsilon^2 L D^2}{2}.$$

Since $\delta < \frac{r}{2}$ and $\varepsilon = 1/N$, we have

$$(1-\varepsilon)^N = (1 - \frac{1}{N})(1-\varepsilon)^{N-1} \geq \frac{1}{2}(1-\varepsilon)^{N-1} \geq \frac{\delta}{r}(1-\varepsilon)^{N-1} \geq \frac{\delta}{r}(1-\varepsilon)^{N-1}.$$

Since $\tilde{F}$ is non-negative and $G$-Lipschitz, this implies that

$$
\begin{aligned}
\mathbb{E}(\tilde{F}(\mathbf{z}_{N+1})) &\geq \frac{\delta}{r}(1-\varepsilon)^{N-1}\mathbb{E}(\tilde{F}(\mathbf{z}_1)) + (1-\varepsilon)^{N-1}(1-\frac{\delta}{r})\tilde{F}(\mathbf{z}_\delta^*) \\
&\quad - 3\varepsilon DQ^{1/2}N^{2/3} - \frac{N\varepsilon^2 LD^2}{2} \\
&= (1-\varepsilon)^{N-1}\tilde{F}(\mathbf{z}_\delta^*) + \frac{\delta}{r}(1-\varepsilon)^{N-1}(\mathbb{E}(\tilde{F}(\mathbf{z}_1)) - \tilde{F}(\mathbf{z}_\delta^*)) \\
&\quad - 3\varepsilon DQ^{1/2}N^{2/3} - \frac{N\varepsilon^2 LD^2}{2} \\
&\geq (1-\varepsilon)^{N-1}\tilde{F}(\mathbf{z}_\delta^*) - \frac{\delta}{r}(1-\varepsilon)^{N-1}GD - 3\varepsilon DQ^{1/2}N^{2/3} - \frac{N\varepsilon^2 LD^2}{2}.
\end{aligned}
$$

After setting $\varepsilon = 1/N$ and using $(1 - 1/N)^{N-1} \geq e^{-1}$, we see that

$$
\begin{aligned}
e^{-1}\tilde{F}(\mathbf{z}_\delta^*) - \mathbb{E}(\tilde{F}(\mathbf{z}_{N+1})) &\leq \frac{3DQ^{1/2}}{N^{1/3}} + \frac{LD^2}{2N} + \frac{\delta}{r}(1-\varepsilon)^{N-1}GD \\
&\leq \frac{3DQ^{1/2}}{N^{1/3}} + \frac{LD^2}{2N} + \frac{\delta}{r}GD.
\end{aligned}
$$

Using the argument presented in Appendix I, i.e. Lemma 3 and Equation 24, we conclude that

$$
e^{-1}F(\mathbf{z}^*) - \mathbb{E}[F(\mathbf{z}_{N+1})] \leq \frac{3DQ^{1/2}}{N^{1/3}} + \frac{LD^2}{2N} + \delta G(2 + \frac{\sqrt{d}+2D}{r}). \qquad \square
$$

## K   Proof of Theorem 1 for monotone maps over general convex sets

*Proof.* Using the fact that $\tilde{F}$ is $L$-smooth, we have

$$
\begin{aligned}
\tilde{F}(\mathbf{z}_{n+1}) - \tilde{F}(\mathbf{z}_n) &\geq \langle \nabla(\tilde{F}|_{\mathcal{L}})(\mathbf{z}_n), \mathbf{z}_{n+1} - \mathbf{z}_n \rangle - \frac{L}{2}\|\mathbf{z}_{n+1} - \mathbf{z}_n\|^2 \\
&= \varepsilon\langle \nabla(\tilde{F}|_{\mathcal{L}})(\mathbf{z}_n), \mathbf{v}_n - \mathbf{z}_n \rangle - \frac{\varepsilon^2 L}{2}\|\mathbf{v}_n - \mathbf{z}_n\|^2 \\
&\geq \varepsilon\langle \nabla(\tilde{F}|_{\mathcal{L}})(\mathbf{z}_n), \mathbf{v}_n - \mathbf{z}_n \rangle - \frac{\varepsilon^2 LD^2}{2} \\
&= \varepsilon\left(\langle \bar{\mathbf{g}}_n, \mathbf{v}_n - \mathbf{z}_n \rangle + \langle \nabla(\tilde{F}|_{\mathcal{L}})(\mathbf{z}_n) + \bar{\mathbf{g}}_n, \mathbf{v}_n - \mathbf{z}_n \rangle\right) - \frac{\varepsilon^2 LD^2}{2}.
\end{aligned} \tag{28}
$$

Let $\mathbf{z}_\delta^* := \operatorname{argmax}_{\mathbf{z} \in \mathcal{K}_\delta} \tilde{F}(z)$. Using the fact that $\langle \bar{\mathbf{g}}_n, \mathbf{v}_n \rangle \geq \langle \bar{\mathbf{g}}_n, \mathbf{z}_\delta^* \rangle$, we have

$$
\begin{aligned}
\langle \bar{\mathbf{g}}_n, \mathbf{v}_n - \mathbf{z}_n \rangle &+ \langle \nabla(\tilde{F}|_{\mathcal{L}})(\mathbf{z}_n) - \bar{\mathbf{g}}_n, \mathbf{v}_n - \mathbf{z}_n \rangle \\
&\geq \langle \bar{\mathbf{g}}_n, \mathbf{z}_\delta^* - \mathbf{z}_n \rangle + \langle \nabla(\tilde{F}|_{\mathcal{L}})(\mathbf{z}_n) - \bar{\mathbf{g}}_n, \mathbf{v}_n - \mathbf{z}_n \rangle \\
&= \langle \nabla(\tilde{F}|_{\mathcal{L}})(\mathbf{z}_n), \mathbf{z}_\delta^* - \mathbf{z}_n \rangle - \langle \nabla(\tilde{F}|_{\mathcal{L}})(\mathbf{z}_n) - \bar{\mathbf{g}}_n, \mathbf{z}_\delta^* - \mathbf{v}_n \rangle.
\end{aligned}
$$

Using the Cauchy-Schwarz inequality, we see that

$$
\langle \nabla(\tilde{F}|_{\mathcal{L}})(\mathbf{z}_n) - \bar{\mathbf{g}}_n, \mathbf{z}_\delta^* - \mathbf{v}_n \rangle \leq \|\nabla(\tilde{F}|_{\mathcal{L}})(\mathbf{z}_n) - \bar{\mathbf{g}}_n\|\|\mathbf{z}_\delta^* - \mathbf{v}_n\| \leq D\|\nabla(\tilde{F}|_{\mathcal{L}})(\mathbf{z}_n) - \bar{\mathbf{g}}_n\|.
$$

Therefore

$$
\begin{aligned}
\langle \bar{\mathbf{g}}_n, \mathbf{v}_n - \mathbf{z}_n \rangle &+ \langle \nabla(\tilde{F}|_{\mathcal{L}})(\mathbf{z}_n) - \bar{\mathbf{g}}_n, \mathbf{v}_n - \mathbf{z}_n \rangle \\
&\geq \langle \nabla(\tilde{F}|_{\mathcal{L}})(\mathbf{z}_n), \mathbf{z}_\delta^* - \mathbf{z}_n \rangle - D\|\nabla(\tilde{F}|_{\mathcal{L}})(\mathbf{z}_n) - \bar{\mathbf{g}}_n\|.
\end{aligned}
$$

Plugging this into 28, we get

$$
\tilde{F}(\mathbf{z}_{n+1}) - \tilde{F}(\mathbf{z}_n) \geq \varepsilon\langle \nabla(\tilde{F}|_{\mathcal{L}})(\mathbf{z}_n), \mathbf{z}_\delta^* - \mathbf{z}_n \rangle - \varepsilon D\|\nabla(\tilde{F}|_{\mathcal{L}})(\mathbf{z}_n) - \bar{\mathbf{g}}_n\| - \frac{\varepsilon^2 LD^2}{2}. \tag{29}
$$

Using Lemma 2 and the fact that $\tilde{F}$ is monotone, we see that

$$
\begin{aligned}
\langle \nabla(\tilde{F}|_{\mathcal{L}})(\mathbf{z}_n), \mathbf{z}_\delta^* - \mathbf{z}_n \rangle &\geq \tilde{F}(\mathbf{z}_\delta^* \vee \mathbf{z}_n) + \tilde{F}(\mathbf{z}_\delta^* \wedge \mathbf{z}_n) - 2\tilde{F}(\mathbf{z}_n) \\
&\geq \tilde{F}(\mathbf{z}_\delta^*) + \tilde{F}(\mathbf{z}_\delta^* \wedge \mathbf{z}_n) - 2\tilde{F}(\mathbf{z}_n) \\
&\geq \tilde{F}(\mathbf{z}_\delta^*) - 2\tilde{F}(\mathbf{z}_n).
\end{aligned}
$$

After plugging this into (29), we get

$$
\tilde{F}(\mathbf{z}_{n+1}) - \tilde{F}(\mathbf{z}_n) \geq \varepsilon\tilde{F}(\mathbf{z}_\delta^*) - 2\varepsilon\tilde{F}(\mathbf{z}_n) - \varepsilon D\|\nabla(\tilde{F}|_{\mathcal{L}})(\mathbf{z}_n) - \bar{\mathbf{g}}_n\| - \frac{\varepsilon^2 L D^2}{2}.
$$

After taking the expectation, using Lemma 9 and re-arranging the terms, we see that

$$
\mathbb{E}[\tilde{F}(\mathbf{z}_{n+1})] \geq (1 - 2\varepsilon)\mathbb{E}[\tilde{F}(\mathbf{z}_n)] + \varepsilon\tilde{F}(\mathbf{z}_\delta^*) - \frac{\varepsilon D Q^{1/2}}{(n+4)^{1/3}} - \frac{\varepsilon^2 L D^2}{2}.
$$

Using this inequality recursively together with Equation (22) and the fact that $\tilde{F}$ is non-negative, we get

$$
\begin{aligned}
\mathbb{E}[\tilde{F}(\mathbf{z}_{N+1})] &\geq (1 - 2\varepsilon)^N \mathbb{E}[\tilde{F}(\mathbf{z}_1)] + \varepsilon\tilde{F}(\mathbf{z}_\delta^*) \sum_{n=1}^{N} (1 - 2\varepsilon)^{N-n} \\
&\qquad - \sum_{n=1}^{N} \frac{\varepsilon D Q^{1/2}}{(n+4)^{1/3}} - \frac{N\varepsilon^2 L D^2}{2}. \\
&\geq \frac{1}{2}(1 - 2\varepsilon)^N \mathbb{E}[\tilde{F}(\mathbf{z}_1)] + \varepsilon\tilde{F}(\mathbf{z}_\delta^*) \sum_{n=1}^{N} (1 - 2\varepsilon)^{N-n} \\
&\qquad - 3\varepsilon D Q^{1/2} N^{2/3} - \frac{N\varepsilon^2 L D^2}{2} \\
&= \frac{1}{2}(1 - 2\varepsilon)^N \mathbb{E}[\tilde{F}(\mathbf{z}_1)] + \frac{1}{2}(1 - (1 - 2\varepsilon)^N)\mathbb{E}[\tilde{F}(\mathbf{z}_\delta^*)] \\
&\qquad - 3\varepsilon D Q^{1/2} N^{2/3} - \frac{N\varepsilon^2 L D^2}{2} \\
&= \frac{1}{2}\tilde{F}(\mathbf{z}_\delta^*) - \frac{1}{2}(1 - 2\varepsilon)^N (\tilde{F}(\mathbf{z}_\delta^*) - \mathbb{E}[\tilde{F}(\mathbf{z}_1)]) \\
&\qquad - 3\varepsilon D Q^{1/2} N^{2/3} - \frac{N\varepsilon^2 L D^2}{2} \\
&\geq \frac{1}{2}\tilde{F}(\mathbf{z}_\delta^*) - \frac{1}{2}(1 - 2\varepsilon)^N DG - 3\varepsilon D Q^{1/2} N^{2/3} - \frac{N\varepsilon^2 L D^2}{2}.
\end{aligned}
$$

Note that $(1 - \log(N)/N)^N \leq e^{-\log(N)} = 1/N$. Therefore, since $\varepsilon = \log(N)/2N$, we have

$$
\mathbb{E}[\tilde{F}(\mathbf{z}_{N+1})] \geq \frac{1}{2}\tilde{F}(\mathbf{z}_\delta^*) - \frac{DG}{2N} - \frac{3DQ^{1/2}\log(N)}{2N^{1/3}} - \frac{LD^2\log(N)^2}{8N}. \tag{30}
$$

According to Lemma 3, we have $\tilde{F}(\mathbf{z}_{N+1}) \leq F(\mathbf{z}_{N+1}) + \delta G$. Moreover, using Lemma 7, we see that $\mathbf{z}^* \in \mathbb{B}_{\delta'}(\mathcal{K}_\delta)$ where $\delta' = \delta D/r$. Therefore, there is a point $\mathbf{y}^* \in \mathcal{K}_\delta$ such that $\|\mathbf{y}^* - \mathbf{z}^*\| \leq \delta'$.

$$
\tilde{F}(\mathbf{z}_\delta^*) \geq \tilde{F}(\mathbf{y}^*) \geq \tilde{F}(\mathbf{y}^*) \geq F(\mathbf{y}^*) - \delta G \geq F(\mathbf{z}^*) - (\delta + \frac{\delta D}{r})G. \tag{31}
$$

After plugging these into (30), we see that

$$
\begin{aligned}
\frac{1}{2}\tilde{F}(\mathbf{z}^*) &- \mathbb{E}[\tilde{F}(\mathbf{z}_{N+1})] \\
&\leq \frac{3DQ^{1/2}\log(N)}{2N^{1/3}} + \frac{4DG + LD^2\log(N)^2}{8N} + \delta G(2 + \frac{D}{r}).
\end{aligned}
$$

which completes the proof. $\qquad\square$

## L    Proof of Theorem 1 for non-monotone maps over general convex sets

*Proof.* First we show that

$$1 - \|\mathbf{z}_n\|_\infty \geq (1 - \varepsilon)^{n-1}(1 - \|\mathbf{z}_1\|_\infty), \tag{32}$$

for all $1 \leq n \leq N + 1$. We use induction on $n$ to show that for each coordinate $1 \leq i \leq d$, we have $1 - [\mathbf{z}_n]_i \geq (1 - \varepsilon)^{n-1}(1 - [\mathbf{z}_1]_i)$. The claim is obvious for $n = 1$. Assuming that the inequality is true for $n$, we have

$$1 - [\mathbf{z}_{n+1}]_i = 1 - (1 - \varepsilon)[\mathbf{z}_n]_i - \varepsilon[\mathbf{v}_n]_i \geq 1 - (1 - \varepsilon)[\mathbf{z}_n]_i - \varepsilon$$
$$= (1 - \varepsilon)(1 - [\mathbf{z}_n]_i) \geq (1 - \varepsilon)^n(1 - [\mathbf{z}_1]_i),$$

which completes the proof by induction.

Let $\mathbf{z}_\delta^* := \mathrm{argmax}_{\mathbf{z} \in \mathcal{K}_\delta} \tilde{F}(z)$. Using the same arguments as in Appendix K, we see that

$$\tilde{F}(\mathbf{z}_{n+1}) - \tilde{F}(\mathbf{z}_n) \geq \varepsilon \langle \nabla(\tilde{F}|_\mathcal{L})(\mathbf{z}_n), \mathbf{z}_\delta^* - \mathbf{z}_n \rangle - \varepsilon D \|\nabla(\tilde{F}|_\mathcal{L})(\mathbf{z}_n) - \bar{\mathbf{g}}_n\| - \frac{\varepsilon^2 L D^2}{2}.$$

Using Lemmas 2 and 1 and Equation (32), we have

$$\langle \nabla(\tilde{F}|_\mathcal{L})(\mathbf{z}_n), \mathbf{z}_\delta^* - \mathbf{z}_n \rangle \geq \tilde{F}(\mathbf{z}_\delta^* \vee \mathbf{z}_n) + \tilde{F}(\mathbf{z}_\delta^* \wedge \mathbf{z}_n) - 2\tilde{F}(\mathbf{z}_n)$$
$$\geq (1 - \|\mathbf{z}_n\|_\infty)\tilde{F}(\mathbf{z}_\delta^*) + \tilde{F}(\mathbf{z}_\delta^* \wedge \mathbf{z}_n) - 2\tilde{F}(\mathbf{z}_n)$$
$$\geq (1 - \varepsilon)^{n-1}(1 - \|\mathbf{z}_1\|_\infty)\tilde{F}(\mathbf{z}_\delta^*) + \tilde{F}(\mathbf{z}_\delta^* \wedge \mathbf{z}_n) - 2\tilde{F}(\mathbf{z}_n)$$
$$\geq (1 - \varepsilon)^{n-1}(1 - \|\mathbf{z}_1\|_\infty)\tilde{F}(\mathbf{z}_\delta^*) - 2\tilde{F}(\mathbf{z}_n).$$

Therefore

$$\tilde{F}(\mathbf{z}_{n+1}) - \tilde{F}(\mathbf{z}_n) \geq \varepsilon(1 - \varepsilon)^{n-1}(1 - \|\mathbf{z}_1\|_\infty)\tilde{F}(\mathbf{z}_\delta^*) - 2\varepsilon\tilde{F}(\mathbf{z}_n)$$
$$-\varepsilon D \|\nabla(\tilde{F}|_\mathcal{L})(\mathbf{z}_n) - \bar{\mathbf{g}}_n\| - \frac{\varepsilon^2 L D^2}{2}.$$

After taking the expectation, using Lemma 9 and re-arranging the terms, we see that

$$\mathbb{E}[\tilde{F}(\mathbf{z}_{n+1})] \geq (1 - 2\varepsilon)\mathbb{E}[\tilde{F}(\mathbf{z}_n)] + \varepsilon(1 - \varepsilon)^{n-1}(1 - \|\mathbf{z}_1\|_\infty)\tilde{F}(\mathbf{z}_\delta^*)$$
$$- \frac{\varepsilon D Q^{1/2}}{(n + 4)^{1/3}} - \frac{\varepsilon^2 L D^2}{2}. \tag{33}$$

Using this inequality recursively together with Equation (22), we see that

$$\mathbb{E}[\tilde{F}(\mathbf{z}_{N+1})] \geq \varepsilon(1 - \|\mathbf{z}_1\|_\infty)\tilde{F}(\mathbf{z}_\delta^*)\sum_{n=1}^{N}(1 - \varepsilon)^{n-1}(1 - 2\varepsilon)^{N-n}$$
$$+ (1 - 2\varepsilon)^N \mathbb{E}[\tilde{F}(\mathbf{z}_1)] - \sum_{n=1}^{N} \frac{\varepsilon D Q^{1/2}}{(n + 4)^{1/3}} - \frac{N\varepsilon^2 L D^2}{2}. \tag{34}$$

Elementary calculations show that $(1 - \frac{c}{N})^{N-1} \geq e^{-c}$ for $0 \leq c \leq 2$ and $N \geq 4$. [1]  Therefore, since $\varepsilon = \log(2)/N$, we have

$$(1 - 2\varepsilon)^N \geq e^{-2\log(2)}(1 - 2\varepsilon) = \frac{1}{4}\left(1 - \frac{2\log(2)}{N}\right) \geq \frac{1}{4N}. \tag{35}$$

On the other hand

$$\varepsilon \sum_{n=1}^{N}(1 - 2\varepsilon)^{N-n}(1 - \varepsilon)^{n-1} = \varepsilon(1 - 2\varepsilon)^{N-1}\sum_{n=1}^{N}\left(\frac{1 - \varepsilon}{1 - 2\varepsilon}\right)^{n-1}$$
$$\geq \varepsilon(1 - 2\varepsilon)^{N-1}\sum_{n=1}^{N}(1 + \varepsilon)^{n-1}$$
$$= (1 - 2\varepsilon)^{N-1}((1 + \varepsilon)^N - 1).$$

---

[1]For $0 \leq x \leq \frac{1}{2}$, we have $\log(1 - x) \geq -x - \frac{x^2}{2} - x^3$. Therefore, for $0 \leq c \leq 2$ and $N \geq 4$, we have $\log(1 - \frac{c}{N}) \geq -\frac{c}{N} - \frac{c^2}{2N^2} - \frac{c^3}{N^3} \geq -\frac{c}{N-1}$.

We have $(1 + \frac{c}{N})^N \geq e^c(1 - \frac{c^2}{2N})$ for $c \geq 0$ and $N \geq 1$. [2] Therefore

$$\varepsilon \sum_{n=1}^{N} (1 - 2\varepsilon)^{N-n}(1 - \varepsilon)^{n-1} = (1 - 2\varepsilon)^{N-1}\left((1 + \varepsilon)^N - 1\right)$$

$$\geq e^{-2\log(2)}\left(\left(1 + \frac{\log(2)}{N}\right)^N - 1\right)$$

$$\geq e^{-2\log(2)}\left(e^{\log 2}\left(1 - \frac{\log(2)^2}{2N}\right) - 1\right)$$

$$= \frac{1}{4}\left(1 - \frac{\log(2)^2}{N}\right)$$

$$\geq \frac{1}{4} - \frac{1}{4N}.$$

Plugging this and 35 into 34 and using the fact that $\tilde{F}(z_1)$ is non-negative, we get

$$\mathbb{E}[\tilde{F}(\mathbf{z}_{N+1})] \geq \left(\frac{1}{4} - \frac{1}{4N}\right)(1 - \|\mathbf{z}_1\|_\infty)\tilde{F}(\mathbf{z}_\delta^*) + \frac{1}{4N}\mathbb{E}[\tilde{F}(\mathbf{z}_1)] - \frac{3DQ^{1/2}}{N^{1/3}} - \frac{LD^2}{2N}$$

$$\geq \frac{1}{4}(1 - \|\mathbf{z}_1\|_\infty)\tilde{F}(\mathbf{z}_\delta^*) + \frac{1}{4N}\left(\mathbb{E}[\tilde{F}(\mathbf{z}_1)] - \tilde{F}(\mathbf{z}_\delta^*)\right) - \frac{3DQ^{1/2}}{N^{1/3}} - \frac{LD^2}{2N}$$

$$\geq \frac{1}{4}(1 - \|\mathbf{z}_1\|_\infty)\tilde{F}(\mathbf{z}_\delta^*) - \frac{3DQ^{1/2}}{N^{1/3}} - \frac{DG + 2LD^2}{4N}.$$

Using the same argument as in Appendix K, we obtain

$$\frac{1}{4}(1 - \|\mathbf{z}_1\|_\infty)F(\mathbf{z}^*) - \mathbb{E}[F(\mathbf{z}_{N+1})] \leq \frac{3DQ^{1/2}}{N^{1/3}} + \frac{DG + 2LD^2}{4N} + \delta G(2 + \frac{D}{r}). \qquad \square$$

## M  Proof of Theorem 2

*Proof.* Let $T = O(BN)$ denote the number of evaluations[3] and let $\mathcal{E}_\alpha := \alpha F(\mathbf{z}^*) - \mathbb{E}[F(\mathbf{z}_{N+1})]$ denote the $\alpha$-approximation error. We prove Cases 1-4 separately. Note that $F$ being non-monotone or $\mathbf{0} \in \mathcal{K}$ correspond to cases (A), (B) and (D) of Theorem 1 where $\log(N)$ does not appear in the approximation error bound, which is why $\tilde{O}$ can be replaced with $O$.

**Case 1 (deterministic gradient oracle):**  In this case, we have $Q = \delta = 0$. According to Theorem 1, in cases (A), (B) and (D), the approximation error is bounded by $\frac{DG+2LD^2}{4N} = O(N^{-1})$, and thus we choose $T = N = \Theta(1/\epsilon)$ to get $\mathcal{E}_\alpha = O(\epsilon)$. Similarly, in case (C), we have

$$\mathcal{E}_\alpha \leq \frac{4DG + LD^2\log(N)^2}{8N} = O(N^{-1}\log(N)^2).$$

We choose $T = N = \Theta(\log^2(\epsilon)/\epsilon)$ to bound $\alpha$-approximation error by $O(\epsilon)$.

**Case 2 (stochastic gradient oracle):**  In this case, we have $Q = \Theta(1)$ and $\delta = 0$. According to Theorem 1, in cases (A), (B) and (D), the approximation error is bounded by

$$\frac{3DQ^{1/2}}{N^{1/3}} + \frac{DG + 2LD^2}{4N} = O(N^{-1/3} + N^{-1}) = O(N^{-1/3}),$$

so we choose $N = \Theta(1/\epsilon^3)$, $B = 1$ and $T = \Theta(1/\epsilon^3)$ to get $\mathcal{E}_\alpha = O(\epsilon)$. Similarly, in case (C), we have

$$\mathcal{E}_\alpha \leq \frac{3DQ^{1/2}\log(N)}{2N^{1/3}} + \frac{4DG + LD^2\log(N)^2}{8N} = O(N^{-1/3}\log(N) + N^{-1}\log(N)^2)$$

---

[2]For $x \geq 0$, we have $\log(1 + x) \geq x - \frac{x^2}{2}$ and $-x \geq \log(1 - x)$. Therefore $N\log(1 + \frac{c}{N}) \geq N(\frac{c}{N} - \frac{c^2}{2N^2}) = c - \frac{c^2}{2N} \geq c + \log(1 - \frac{c^2}{2N})$.

[3]We have $T = BN$ when we have access to a gradient oracle and $T = 2BN$ otherwise.

Since $\mathcal{E}_\alpha \leq O(N^{-1/3} \log(N)^2)$, we choose $N = \Theta(\log^6(\epsilon)/\epsilon^3)$, $B = 1$ and $T = \Theta(\log^6(\epsilon)/\epsilon^3)$ to bound $\alpha$-approximation error by $O(\epsilon)$.

**Case 3 (deterministic value oracle):** In this case, we have $Q = \Theta(1)$ and $\delta \neq 0$. According to Theorem 1, in cases (A), (B) and (D), the approximation error is bounded by

$$\frac{3DQ^{1/2}}{N^{1/3}} + \frac{DG + 2LD^2}{4N} + O(\delta) = O(N^{-1/3} + \delta),$$

so we choose $\delta = \Theta(\epsilon)$, $N = \Theta(1/\epsilon^3)$, $B = 1$ and $T = \Theta(1/\epsilon^3)$ to get $\mathcal{E}_\alpha = O(\epsilon)$. Similarly, in case (C), we have

$$\mathcal{E}_\alpha \leq \frac{3DQ^{1/2} \log(N)}{2N^{1/3}} + \frac{4DG + LD^2 \log(N)^2}{8N} + O(\delta) = O(N^{-1/3} \log(N)^2 + \delta).$$

We choose $\delta = \Theta(\epsilon)$, $N = \Theta(\log^6(\epsilon)/\epsilon^3)$, $B = 1$ and $T = \Theta(\log^6(\epsilon)/\epsilon^3)$ to bound $\alpha$-approximation error by $O(\epsilon)$.

**Case 4 (stochastic value oracle):** In this case, we have $Q = O(1) + O(\frac{1}{\delta^2 B})$ and $\delta \neq 0$. According to Theorem 1, in cases (A), (B) and (D), the approximation error is bounded by

$$\frac{3DQ^{1/2}}{N^{1/3}} + \frac{DG + 2LD^2}{4N} + O(\delta) = O(Q^{1/2} N^{-1/3} + N^{-1} + \delta)$$
$$= O(N^{-1/3} + \delta^{-1} B^{-1/2} N^{-1/3} + \delta),$$

so we choose $\delta = \Theta(\epsilon)$, $N = \Theta(1/\epsilon^3)$, $B = \Theta(1/\epsilon^2)$ and $T = \Theta(1/\epsilon^5)$ to get $\mathcal{E}_\alpha = O(\epsilon)$. Similarly, in case (C), we have

$$\mathcal{E}_\alpha \leq \frac{3DQ^{1/2} \log(N)}{2N^{1/3}} + \frac{4DG + LD^2 \log(N)^2}{8N} + O(\delta)$$
$$= O(Q^{1/2} N^{-1/3} \log(N) + N^{-1} \log(N)^2 + \delta)$$
$$= O(N^{-1/3} \log(N)^2 + \delta^{-1} B^{-1/2} N^{-1/3} \log(N)^2 + \delta).$$

We choose $\delta = \Theta(\epsilon)$, $N = \Theta(\log^6(\epsilon)/\epsilon^3)$, $B = \Theta(1/\epsilon^2)$, and $T = \Theta(\log^6(\epsilon)/\epsilon^5)$ to bound $\alpha$-approximation error by $O(\epsilon)$. $\qquad\square$

## N   Proof of Theorem 3

*Proof.* Since the parameters of Algorithm 2 are chosen according to Theorem 2, we see that the $\alpha$-approximation error is bounded by $\tilde{O}(T_0^{-\beta})$ where $\beta = 1/3$ in case 2 (stochastic gradient oracle) and $\beta = 1/5$ in case 4 (stochastic value oracle).

Recall that $F$ is $G$-Lipschitz and the feasible region $\mathcal{K}$ has diameter $D$. Thus, during the first $T_0$ time-steps, the $\alpha$-regret can be bounded by

$$\sup_{z,z' \in \mathcal{K}} \alpha F(\mathbf{z}) - F(\mathbf{z}') \leq \sup_{z,z' \in \mathcal{K}} F(\mathbf{z}) - F(\mathbf{z}') \leq DG.$$

Therefore the total $\alpha$-regret is bounded by

$$T_0 DG + (T - T_0) \tilde{O}(T_0^{-\beta}) \leq T_0 DG + T \tilde{O}(T_0^{-\beta}).$$

Since we have $T_0 = \Theta(T^{\frac{1}{\beta+1}})$, we see that

$$T_0 DG + T \tilde{O}(T_0^{-\beta}) = \tilde{O}(T^{\frac{1}{\beta+1}}) = \begin{cases} \tilde{O}(T^{\frac{3}{4}}) & \text{Case 2,} \\ \tilde{O}(T^{\frac{5}{6}}) & \text{Case 4.} \end{cases}$$

If $F$ is non-monotone or $\mathbf{0} \in \mathcal{K}$, the exact same argument applies with $\tilde{O}$ replaced by $O$. $\qquad\square$