# OpenReview forum: "A Unified Approach for Maximizing Continuous DR-submodular Functions"
_NeurIPS.cc/2023/Conference — NeurIPS 2023 poster_

### Official Review · Reviewer_Dxab · 2023-06-29

**Soundness:** 2 fair
**Presentation:** 3 good
**Contribution:** 2 fair
**Rating:** 3
**Confidence:** 5

**Summary:**

In this work, the authors present a framework for maximizing continuous DR-submodular functions over a range of settings and Oracle access types. To achieve this, they employ a variant of the Frank-Wolfe algorithm which yields either the first guarantees in some cases or comparable results to the SOTA in other cases. The paper is generally well-written, and I have reviewed and found most of the proofs to be accurate. However, I do have a few significant concerns that I would like to highlight as weaknesses.

**Strengths:**

**Major:**
1. This paper presents the first regret analysis with bandit feedback for stochastic DR-submodular functions maximization.

**Minor:**
1. Reducing computational complexity by avoiding projections in two cases.
2. Obtaining the first results on offline DR-submodular maximization over general convex sets and down-closed convex sets.

**Weaknesses:**

**Major concerns:**

1. This paper lacks empirical evaluation, and it is very crucial since the main contribution of this work is on avoiding computationally expensive projections.
2. Motivation. In my opinion, this work lacks a very important paragraph/subsection on the importance of the setting where the oracle provides access to only the function value rather than the gradient of the function since the main results in this work are on this setting. For example, provide some applications in this setting.



**Questions:**

1. Can you please describe the main novelty of this framework?
2. Besides the Online Stochastic DR-submodular maximization with Bandit Feedback result, the other results of this paper seem to be a straightforward extension of prior works. The authors need to highlight the challenges faced when proving these more general results and specify how they managed to overcome the challenges.

**Limitations:**

Yes. The authors adequately addressed the limitations.

---

> ### Author Rebuttal · Authors · 2023-08-10
>
> Thank you for your time in reviewing our paper.
>
> **Weaknesses**
>
> 1. For this concern, there are two points we would like to highlight.
> (1) First, we would like to gently push back on the statement mentioned in the concern that "the main contribution of this work is on avoiding computationally expensive projections."
> Avoiding potentially computationally expensive projections did inspire our choice of using Frank-Wolfe type methods, similar to most SOTA works for offline and online constrained DR-submodular optimization.
> However, for only 3 of the 16 offline problems was a SOTA achieved by a gradient method. Specifically, for monotone functions over general convex set with stochastic/deterministic gradient oracle, in [16], and monotone functions over a convex set containing the origin with access to a stochastic gradient oracle, in "Stochastic continuous submodular aximization: Boosting via non-oblivious function" by Zhang et al. ICML 2022 mentioned by Reviewer 6fHN. Note that in one of these cases, namely the one with deterministic gradient oracle in [16], the complexity is the same as the Frank-Wolfe methods up to logartithmic factors. Likewise, for just 2 of the online stochastic problem settings (one row in Table 2 and the Zhang et al. paper mentioned above) were the SOTA a gradient based method.
> For those problem settings in offline and online case, the sample complexity (respectively regret bound) is worse for our Frank-Wolfe type algorithm compared to the gradient based method.  That was why we did not highlight our method in the corresponding rows.
> (2) Second, we note that in all cases where our result matches SOTA, our unifying framework is a strict generalization of the prior works which focused on specific cases (namely, those with a (*) sign in Table 1). Hence, in all those cases, our algorithm reduces to the SOTA algorithm and therefore all of the experiments done in those papers apply to our algorithm as well.
>
>
> 2. We agree that the introduction should be strengthened as you suggest.
> **Motivation for (feasible) value oracle queries:**  We will revise the introduction to better motivate the importance of developing optimization methods for value oracle queries, including just over the feasible region.  We first highlight two points and then discuss application motivations.
> (1) *Offline-to-online adaptations*
> For online optimization problems, when only bandit feedback is available (it is typically a strong assumption that semi-bandit or full-information feedback is available), then the agent must be able to learn from stochastic value oracle queries over the feasible actions action. By designing offline algorithms that only query feasible points, we made it possible to convert those offline algorithms into online algorithms.  In fact, because of how we designed the offline algorithms, we are able to access them in a black-box fashion for online problems when only bandit feedback is available.
> (2) *More precise characterizations of inherent challenges underlying approximation guarantees*
> As noted above, in developing a unifying framework where we took care to characterize how powerful the oracles were, we identified the underlying causes for an approximation gap between gradient ascent and Frank-Wolfe methods.
> **Applications:**
> We will revise the paper by discussing "classic" example applications that prior works (like [arXiv:2006.13474]) have shown to be instances of constrained DR-submodular maximization, such as influence/revenue maximization, facility location, and non-convex/non-concave quadratic programming, as well as more recently identified applications like serving heterogeneous learners under networking constraints [arXiv:2201.04830] and joint optimization of routing and caching in networks [arXiv:2302.02508]. We will comment on how strong/mild assuming availability of anything more powerful than a value oracle over the feasible region is. For many problems, the ability to evaluate gradients directly requires strong assumptions about problem-specific parameters. We will briefly mention the application examples in the introduction and elaborate in the appendices.
> For example:
> Influence maximization and profit maximization form a family of problems that model choosing advertising resource allocations to maximize the expected number of customers, where there is an underlying diffusion model for how advertising resources spent (stochastically) activate customers over a social network.  For common diffusion models, the objective function is known to be DR-submodular (see for instance [arXiv:2006.13474] or [arXiv:2212.06646]). The revenue (expected number of activated customers) is a monotone objective function; total profit (revenue from activated customers minus advertising costs) is a non-monotone objective. Budget limits are typically modeled as linear constraints.
> One significant challenge with these problems is that the objective function (and the gradients) cannot be analytically evaluated for general (non-bipartite) networks, *even if all the underlying diffusion model parameters are known exactly*.
> The mildest assumptions on knowledge/observability of the network diffusions for offline variants (respectively actions for online variants), especially fitting for user privacy and/or third-party access, leads to instantiations of queries as the agent selecting an advertising allocation within the budget (i.e. feasible point) and observing a (stochastic) count of activated customers. This corresponds to stochastic value oracle queries over the feasible region (respectively bandit feedback for online variants).
>
> **Questions**
>
> We have included a discussion on novelty and significance in a general response box above. While the online stochastic setting is a novelty of our work, we have not listed it as one of the main technical novelties in here or in the submitted article since it is a relatively simple extension of the main results.

---

> > ### Author Response · Authors · 2023-08-19
> >
> > Dear Reviewer Dxab,
> >
> > We wanted to ensure that our comments have adequately addressed your concerns. If you require further clarification or have any additional questions, please don't hesitate to reach out. We appreciate your time and effort in reviewing our paper. Thank you once again.
> >
> > Sincerely,
> >
> > Authors.

---

> > > ### Comment · Reviewer_Dxab · 2023-08-21
> > >
> > > I would like to thank the authors for the detailed response. However, my initial understanding of the paper would appear to be correct. While I do acknowledge the paper's coherent and integrated approach, I find the novelty to be lacking, and the absence of empirical evaluation is notable. To finalize my review, I'll need further discussion with other reviewers. I don't have any further questions for the author-reviewer discussion period.

---

> > > > ### Author Response · Authors · 2023-08-21
> > > >
> > > > Thank you for following up.   We note that the main technical novelties of the paper include:
> > > >
> > > > 1. **Our procedure is the first Frank-Wolfe type algorithm for analyzing monotone functions over general a convex set when the oracle is only allowed to query within the feasible set, for any type of oracle for the objective function (exact/stochastic value/gradient), even an exact gradient oracle.**
> > > >
> > > > 2. **A new construction procedure of a shrunk constraint set that allows us to work with lower dimensional feasible sets when given a value oracle, resulting in the first results on general lower dimensional feasible sets given a value oracle.**
> > > >
> > > > 3. To address challenging problems where only oracles for feasible points are allowed (such as influence and revenue maximization over general networks where any action must be within budget, as described in the previous reply), we introduce the new problem of DR-submodular maximization where the oracle is only available over the feasible set. While we did obtain algorithms and results for the bandit feedback setting, we note that **the main technical contribution here is that it sheds light on a previously unexplained gap in approximation guarantees for monotone DR-submodular maximization.** In other words, we reformulate problems considered before in a way that includes almost all previously considered cases and (in our opinion) is more mathematically coherent.
> > > > To the best of our knowledge, in every paper where the $1/2$ approximation coefficient mentioned for gradient ascent and the $1-1/e$ approximation coefficient for Frank-Wolfe in the monotone setting are compared, the comparison was unwittingly between problems that are inherently mathematically different: [16] and [8] in experiments and main text; [7] and [9] in experiments; [30], [23], and [13] in related work section, [22] in the introduction and Table 2, ”Stochastic continuous submodular Maximization: Boosting via non-oblivious function” by Zhang et al. ICML 2022 in the main claim; and ”Fast First-Order Methods for Monotone Strongly DR-Submodular Maximization.” Fazel et al. (ACDA23), 2023 in the main claims.
> > > >
> > > >
> > > > We hope that the reviewer sees the above as key challenges, since they are not straightforward from the prior works. These novel points are also expanded in the general comments to all reviewers in our previous reply.
> > > >
> > > >
> > > > Regarding experiments,
> > > >
> > > > - We note that for all the cases where our general algorithm specializes to one proposed in prior work (each (*) sign in Table 1, i.e., [3], [4], [11], [22]), all the empirical evaluations in those works using those algorithms hold for our algorithm as well.
> > > >
> > > > - We also note that when only a value oracle over a feasible general convex set is available, there are no baselines to compare with. (Running algorithms using different oracle types would not be as meaningful due to inherently different sample complexities.)
> > > >
> > > > - The focus of the paper is on the theoretical results.

---

### Official Review · Reviewer_f6HN · 2023-07-04

**Soundness:** 4 excellent
**Presentation:** 3 good
**Contribution:** 3 good
**Rating:** 6
**Confidence:** 5

**Summary:**

This paper studies offline constrained DR-submodular maximization in 16 different settings (monotone/non-monotone, down-closed/general convex constraint, gradient/value oracle access, and exact/stochastic oracle) and provides a unified approach to solve all 16 cases with the same Frank-Wolfe algorithmic framework. Moreover, the authors extend their tools to study the online stochastic DR-submodular maximization problem under bandit and semi-bandit feedback models. The provided offline and online algorithms either match or improve the state-of-the-art results for each of the various settings.

**Strengths:**

- Constrained DR-submodular maximization has been studied in numerous works under different assumptions and settings and a number of algorithms have been proposed. In contrast, the algorithmic framework proposed in this paper is general and could be applied to any of the possible settings.
- The paper is well-written and while the proofs are moved to the appendix, the main concepts and ideas are highlighted clearly in the paper.
- Algorithm 1 (BBGE) for gradient estimation contains some novel ideas.

**Weaknesses:**

- While the unifying approach of the paper is interesting, most of the tools and techniques for gradient estimation and the Frank-Wolfe-type algorithm used here have been previously introduced and are not novel.
- The main contributions of this work are the improved complexity (offline setting) and regret bounds (online setting) and these improvements could be highlighted via numerical examples, however, the paper lacks any experiments.

**Questions:**

- I noticed that the following paper is missing from your references. In this work, the authors propose a boosting gradient ascent method that improves the $\frac{1}{2}$ approximation ratio of gradient ascent to $1-\frac{1}{e}$ for the monotone setting. How do their techniques and results compare to yours?

Zhang, Q., Deng, Z., Chen, Z., Hu, H. and Yang, Y., 2022, June. Stochastic continuous submodular maximization: Boosting via non-oblivious function. In International Conference on Machine Learning (pp. 26116-26134). PMLR.

- Is it possible to use your unified approach and provide an analysis for the setting with bounded curvature (similar to what the following paper has done)?

Fazel, Maryam, and Omid Sadeghi. "Fast First-Order Methods for Monotone Strongly DR-Submodular Maximization." In SIAM Conference on Applied and Computational Discrete Algorithms (ACDA23), pp. 169-179. Society for Industrial and Applied Mathematics, 2023.

-  In the online setting with bandit feedback, how many value oracle queries are necessary per iteration to run Algorithm 3?

--------------------------------
I've read the authors' rebuttal, thanks for addressing my questions and concerns.

**Limitations:**

This is a theoretical work and a discussion of potential negative societal impact is not necessary.

---

> ### Author Rebuttal · Authors · 2023-08-10
>
> Thank you for your time and effort in reviewing our submission. We reply to your questions in order below. We also include a discussion on novelty and significance in a general rebuttal box above.
>
> 1. Thank you for pointing this out.
> We first discuss the technique.
> The technique used in that paper is a combination of a novel line integral, referred to as boosting, and the projected gradient ascent.
> The boosting method uses a line integral over the line segment connecting the origin to any point $z$ in the constraint set. (see Theorem 2 part ii)
> In other words, it is working with the assumption that we are allowed to query the oracle on the convex hull of $\mathcal{K} \cup \{0\}$. (outside $\mathcal{K}$)
> Hence, technically speaking, they are not improving $1/2$ to $1-1/e$ within the same problem space, since $1/2$ solves monotone submodular maximization over a general convex set $\mathcal{K}$ where we are only allowed to sample within $\mathcal{K}$.
> Next, we comment on the results. Briefly, they would beat our method and the SOTA included in Table 1 for one of the sixteen settings -- optimizing a monotone function $F$ over convex sets that contain the origin $(0\in \mathcal{K})$, using a stochastic gradient oracle $\nabla F$.  For that problem,  Zhang et al. achieves an approximation of $1-1/e$ with $O(1/\epsilon^2)$ sample complexity.  Our method and the prior SOTA [22] achieved an approximation of $1-1/e$ with $O(1/\epsilon^3)$ sample complexity. We remark that our method, [4], and [22] are Frank-Wolfe type methods and use a linear maximization oracle as a subroutine while Zhang et al. use a quadratic maximization oracle as a subroutine, which for some problems could have high computational complexity.
> Similarly, in Table 2, the results of Zhang et al. should be included in the same category as [7] and [29] and it will be the SOTA with ($1-1/e$)-regret of $O(T^{1/2})$. However, we again note that they use a projection based method, so our result would be SOTA among projection-free algorithms.
>
> 2. Thank you for pointing this out. This is an interesting direction.  It is not immediately clear how the update rule, specifically the value of $v_k$ in Algorithm 3.1. SDRFW in Fazel et al., should change to adapt to other settings as we have studied in order to exploit curvature and strong DR-submodularity to obtain better guarantees. However, we would not be surprised if a unified approach similar in spirit to ours could be applied in the bounded curvature setting and believe it is worth future investigation.
>
> 3. For the online setting with bandit feedback, for each time step in the online problem only a single value oracle query is performed (corresponding to a single action taken at each time step and only the stochastic reward being revealed to the agent). The exploration horizon $T_0$ in Algorithm 3 (for bandit feedback we have $T_0 \gets \lceil T^{5/6} \rceil$) is input as the total number of iterations $N$ in Algorithm 2 when the latter is invoked as a black-box subroutine for exploration (so $N \gets \lceil T^{5/6} \rceil$).

---

> > ### Comment · Reviewer_f6HN · 2023-08-15
> >
> > Thanks for your detailed response. Regarding question 2, if we put aside the idea of strong DR submodularity (and focus only on bounded curvature), the following work provides an optimal and efficient algorithm for submodular maximization with bounded curvature. Is it possible to extend your unified analysis using this work? What are the challenges for this extension?
> >
> > Feldman, Moran. "Guess free maximization of submodular and linear sums." Algorithmica 83.3 (2021): 853-878.

---

> > > ### Author Response · Authors · 2023-08-17
> > >
> > > Thank you for following up and pointing out that work by Feldman. To answer the comment, we note the following:
> > >
> > > 1. For DR-submodular setting, we begin by noting that Algorithm 1 in the previous paper you mentioned [Fazel et al.], may be considered an adaptation of Algorithm 1 of the Feldman paper (which considers set submodular functions) to the monotone DR-submodular setting where the constraint set contains the origin $0 \in \mathcal{K}$.
> > > As we mentioned in our previous reply, any extension of the Fazel paper (with curvature and strong DR-submodularity) to settings with non-monotone functions or when the constraint set does not contains the origin, would require a non-trivial change in the update rule. However, we believe such extension to be possible.
> > > A natural starting point  for monotone DR-submodular functions with (just) bounded curvature would be to use Fazel et al. paper by "removing" the strong DR-submodularity. However, doing so is not trivial (if even possible).
> > > Setting $\mu \gets 0$ simplifies the update formula $v_k$ in SDRFW algorithm, however, their main result is only applicable when $\mu > 0$. More precisely, we note that the approximation bounds depend on a number of iterations $K \propto 1/\mu$, which is meaningless for $\mu=0$, so we would need to modify the proof, and possibly the update rule, to obtain a result using this approach.
> > > Another significant issue is with the algorithm design, as Fazel et al. algorithm requires as input a particular linear function $\ell_i = \min_x \nabla_i f(x)$ which in general may be challenging to compute.  The function essentially finds, for each coordinate, worst case marginal gains achieved in the feasible region.  It is not clear to us how we could compute that efficiently (even with exact gradient oracles) to begin with in addition to the FW steps that incorporate that. Further, with stochastic gradient estimates (via a stochastic gradient oracle or using samples from a value oracle), it is not immediately clear how robust the algorithm would be to an inexact $\ell_i$.
> > > We remark that Fazel et al.'s gradient ascent algorithm (Algorithm 2) interestingly does not require knowledge of $\mu$ or require as input the same linear function (constructed from $F$) as their FW type algorithm SDRFW did. That result looks more promising to generalize to different feasible regions and objective oracle types. The approximation coefficient obtained $1/(1+c_f)$. Note that this result is for monotone DR-submodular functions over general convex sets and therefore their approximation coefficient is $1/2$ rather than $1-1/e$ when $c = 1$.
> > >
> > > 2. The mentioned paper (Feldman, "Guess free$\dots$") does not discuss maximization of submodular functions with bounded curvature. It considers the closely related problem of optimizing the sum of a submodular function and a linear function, which is used as a subroutine for bounded curvature (DR-)submodular maximization as in [Sviridenko et al., Mathematics of Operations Research 42.4 (2017): 1197-1218. https://pubsonline.informs.org/doi/abs/10.1287/moor.2016.0842 ] or [Fazel et al.], and will be discussed in (3) below. For a given known linear function (i.e., separate value/gradient oracles for the DR-submodular and the linear functions), we think it is plausible can likely extend our paper to the sum of a DR-submodular function and a linear function. However, a detailed investigation is needed, since the approximation ratios need to be seen if they still hold (where approximation ratio is in the front of only DR-submodular function). Further, the update rules would need modifications. We believe that a careful analysis might work out, and is a good topic for the future work.
> > >
> > > 3. As mentioned above, optimizing the sum of a submodular function and a linear function is used as a subroutine for bounded curvature (DR-)submodular maximization.
> > > In other words, the first step for maximizing a bounded curvature (DR-)submodular function is to decompose it as a sum of a linear and a (DR-)submodular function.
> > > [Sviridenko et al., 2017] constructs a linear function for the discrete case, which can be evaluated using multiple oracle calls. However, such an approach is shown only for monotone (set) functions. Further, in order to obtain this linear function, we may need to query outside the feasible set. For the DR-submodular case, a linear function construction is given in Fazel et al., as mentioned in point (1) above, whose computation can be challenging and it is not evident how to compute it efficiently with only oracle calls.
> > >
> > > In summary, we believe that the problem of extending our work on the direction of bounded curvature is an interesting problem. However, we do not believe this to be an easy direction. Nevertheless, we will point this possible extension in the future works section in the final version.

---

> > > > ### Comment · Reviewer_f6HN · 2023-08-18
> > > >
> > > > Thanks, I see your point. I have no further questions.

---

### Official Review · Reviewer_L9XA · 2023-07-06

**Soundness:** 4 excellent
**Presentation:** 4 excellent
**Contribution:** 3 good
**Rating:** 7
**Confidence:** 3

**Summary:**

The paper studies maximizing stochastic DR-submodular functions under 16 different settings, which depend on 1) whether the function is monotone or not, 2) the feasible region is a downward-closed or a general convex set, 3) gradient or value oracle access is available, and 4) or the oracle is exact or stochastic. The authors present a unified approach based on the Frank-Wolfe meta-algorithm, which 1) provides the first oracle complexity guarantees in 9 settings, 2) reduces the computational complexity by avoiding projections in two settings, and 3) matches guarantees in the remaining 5 settings. The paper also considers online versions of the problem with bandit feedback and semi-bandit feedback and presents online algorithms with improved regret bounds.

**Strengths:**

1. The paper provides a unified approach for stochastic DR-submodular maximization, which encompasses a range of settings and beats or matches the SOTA results.

2. The paper presents the first regret analysis for online stochastic DR-submodular maximization with bandit feedback.

3. Technically, a novel construction procedure of a shrunk constraint set is invented that allows us to work with lower dimensional feasible sets when given a value oracle.

4. The paper is organized very well, making it read clearly although there are so many settings. Besides, no typo was found after I read it.

**Weaknesses:**

1. I notice that only in the case where the value oracle is available, new guarantees that beat the SOTA are provided. So I consider it as the main part of the paper's contributions. In this case, the paper uses many standard techniques like the smoothing trick, the two-point estimator (for estimating gradients), and the momentum technique (for stochastic oracle). And to me what makes the paper different is the construction of a shrunk constraint set. But without reading the appendix, I can not figure out why such a construction is introduced and how it can achieve the claimed guarantees. So, I can not determine whether the paper enjoys excellent technical novelty or just consists of a refined combination of known techniques.

**Questions:**

1. If possible, please provides more intuition in the main text about what difficulties the construction of a shrunk constraint set is used to solve and how it can solve these difficulties.

2. Very recently, There is a paper titled "Bandit Multi-linear DR-Submodular Maximization and Its Applications on Adversarial Submodular Bandits" [arXiv: 2305.12402, ICML23], which presents a $\tilde{O}(T^{3/4})$ regret for monotone submodular maximization with adversary bandit feedback. This partially beats your results for bandit submodular maximization. As the approaches are different and the arXiv paper was unavailable during the NeurIPS submission, I do not think it is a "weakness". But it is suggested to include the paper in the Related Work.

**Limitations:**

Yes, as stated in the paper, any non-trivial lower bound for the problem would be exciting.

---

> ### Author Rebuttal · Authors · 2023-08-10
>
> Thank you for time and efforts in reviewing our submission.  Below we first provide a response to the points you raised in the "Weaknesses" section and then reply to your questions in order.
>
> **Weaknesses**
>
> 1. In the "global rebuttal" above, we explain the novelty of our work in more detail and reply to your question about the shrunken set construction below.
>
> **Questions**
>
> 1. The original version of one-point gradient estimator introduced in [14] is a technique that is used extensively [17, 1, 26, 29, 9, 30] in the literature to estimate the gradient of a function using a single sample. To use this technique for estimating $\nabla F(z)$, one needs to sample points in a sphere centered at $z$. Therefore, it is only applicable when the function can be queried over an open set in $\mathbb{R}^d$. In that case, the original constraint set $\mathcal{K}$ should be shrunk to get $\mathcal{K}'$ so that an open neighbourhood of $\mathcal{K}'$ is contained in $\mathcal{K}$. An example of such construction is done in Appendix D in [29], when $\mathcal{K}$ is an open downward closed convex set.
> When the constraint set is a lower dimensional convex set in $\mathbb{R}^d$, the technique needs to be modified since no open neighbourhood of $\mathcal{K}'$ is contained in $\mathcal{K}$.
> In Lemma 5, we generalize the one-point gradient estimator from [14] so that it does not require sampling from a sphere. This allows us to propose a new construction that is much simpler than [29] and works for general convex sets.
>
> 2. Thank you for bringing this new paper to our attention.  It is exciting to see how active this area is.
> We will discuss this paper in the related work section. In Table 3 (line 553), we include results for stochastic and adversarial online DR-submodular maximization under bandit feedback. This new paper by Wan et al. belongs to the same category as the reference [29], i.e. in the second row for monotone objective functions with a constraint set that contains the origin ($0 \in \mathcal{K}$), which improves the adversarial setting regret bound from $O(T^{8/9})$ to $O(T^{3/4})$.
> However, it should be noted that their algorithm relies on a ``self-concordant barrier'' $\Phi(x)$ of the constraint set which increases the computational complexity of the algorithm.
> More precisely, in Algorithm 1, line 13, of that paper we see that in each iteration a constrained optimization problem needs to be solved as a subroutine, for which the objective function is convex, but is not linear (like in our Frank-Wolfe type algorithm) or even quadratic (like in projected gradient ascent works like [16]).
> Thus, for some problems, the computational complexity could be significantly higher than even gradient ascent.
> It is also important to note that the guarantees depend on the existence of a self-concordant barrier. In Appendix D of their paper, they discuss the construction of self-concordant barriers when the constraint set is a product of simplices. It is unclear to us how  computationally expensive it is to  construct a self-concordant barrier for a general convex set containing the origin.
> We note that the ICML paper references Niazadeh et al. 2021 ([24] in our paper) as the previous SOTA for monotone online adversarial  with bandit feedback with $O(T^{5/6})$. However, in our submission in Appendix B (lines 618-627), we point out an error in their analysis (due to using stochastic gradient samples for a subroutine designed for exact gradients, without controlling for the subsequent variance such as with momentum).
> Hence, in the paper we will state that the SOTA using only a linear maximization oracle as a subroutine is $O(T^{8/9})$ and the SOTA without being restricted to using linear oracles is $O(T^{3/4})$.

---

> > ### Comment · Reviewer_L9XA · 2023-08-12
> >
> > Thanks for your reply. I read all reviews, rebuttals, and comments and now I have a better understanding of the paper's contributions and technical novelties. I'm looking forward to reading the final version of this paper.

---

> > > ### Author Response · Authors · 2023-08-12
> > >
> > > Thank you very much for your time and your comments!

---

### Official Review · Reviewer_GwvC · 2023-07-06

**Soundness:** 2 fair
**Presentation:** 3 good
**Contribution:** 3 good
**Rating:** 6
**Confidence:** 3

**Summary:**

This paper considers the problem of maximizing a continuous DR-submodular function over a convex set, under various settings for the objective (monotone/non-monotone), constraint set (downward-closed/includes the origin/general set), and oracle access types (deterministic/stochastic gradient/value oracle) -- 16 settings in total. For any type of oracle, the oracle is only allowed to query within the feasible set. The authors propose a Frank-Wolfe meta-algorithm and its different variants for each of the considered settings which achieve the best known approximation guarantee in each case. These approximation guarantees are the first ones in nine of the considered settings, reduce the computational complexity in two cases, and match existing guarantees in the remaining five cases. The authors also extend these results to the online stochastic setting with bandit and semi-bandit feedback. They provide the first regret bounds for the bandit setting, and in the semi-bandit setting, they provide the first regret bound in one case, improve over exiting bounds in two cases, and improve the computational complexity in one case.

**Strengths:**


- The paper provide the first approximation guarantees for various settings of DR-submodular maximization, where the oracle is only allowed to query within the feasible set, and improve over existing results in few other settings.
- The paper cover a wide range of settings under a unified algorithm and analysis.
- The presentation of the main paper is very good, especially taking into consideration the amount of material covered.
- Relation to related work is clear: the authors discuss how their results compare to existing ones, and which ideas/proofs are based on existing work.


**Weaknesses:**


- Potential error in Lemma 5 (see questions), which affects all results with the function value oracle. If incorrect, this is easily fixable but the algorithm would require knowing the dimension of the affine hull of the constraint set.

- The presentation in the appendix can be improved to make it easier to read the proofs (see suggestions in questions).

**Questions:**

- In Lemma 5, shouldn't d be replaced by k=dim(A) inside the expectation? Applying Lemma 4 with restriction to the first k coordinates is equivalent to applying it with sampling from a sphere of dimension k no?

- Why is the claim on lines 671-672 in Lemma 5 true? Please explain or provide a reference.

- The notations $\hat{G}(z)|\_\mathcal{L}$ and $\tilde{F}|\_\mathcal{L}$ are not defined, which makes it hard to understand some of the proofs. Are these the projection of $\hat{G}$ on $\mathcal{L}$ and the restriction of the domain of $\tilde{F}$ to $\mathcal{L}$?

- $\mathcal{L}$ is defined as $\mathrm{aff}(K)$ in Lemma 9 and the proof of Theorem 1, but in Algorithm 2 it is defined as $\mathrm{aff}(K) - z_1$.

- I recommend motivating why it is important to restrict the oracle to query only feasible points.

- It would be good to also include known lower bounds for settings where they are available.

- h is not defined in Table 1.

- In the appendix, it is helpful to remind the reader of what different terms refer to or to have a table listing them.

**Limitations:**

yes

---

> ### Author Rebuttal · Authors · 2023-08-10
>
> Thank you for your careful reading and helpful suggestions.  Below we reply to your questions in order.
>
> 1. You are correct.  In Lemma 5, the ratio $d/2\delta$ in the expectation should be replaced by $k/2\delta$ where $k = dim(A)$.
> Similarly, in Algorithm 1, the ratio $d/2\delta$ used in line 4 should be replaced with $k/2\delta$.
> With these modifications, the algorithms and proofs would be correct as is.
>
> 2. For reference, that line is *Note that the function $F$ is defined only on $\mathcal{D}$ and therefore the gradient $\nabla F$ lives within the linear space $A$.*  Preceding that we defined $A$ as the affine hull of $\mathcal{D}$, i.e. $A:= \text{aff}(D).$
> We will include the following explanation in a footnote to clarify the statement.
> Let $f : M \to \mathbb{R}$ be a differentiable function where $M$ is a manifold in $\mathbb{R}^n$ for some choice of $n \geq 1$.
> For each $z \in M$, the total derivative of $f$, is a linear function $D f(z) : T_z(M) \to \mathbb{R}$ from tangent space of $M$ at the point $z$ to $\mathbb{R}$.
> Then gradient of $f$, i.e. $\nabla f$ is the vector field on $M$ for which we have $\langle \nabla f(z), v \rangle = (D f(z))(v)$, for all $z \in M$ and $v \in T_z(M)$.
> In particular, since it is a vector field, at each point $z \in M$, the value of $\nabla f(z)$ is a vector in the tangent space $T_z(M) \subseteq \mathbb{R}^n$.
> As a special case, if $M \subseteq A$ where $A$ is an affine space, then the tangent space $T_z(M) \subseteq A$ for all $z \in M$ and therefore $\nabla F(z) \subseteq A$ for all $z \in M$.
>
> 3. For reference, that notation appears in Lemma 9 (starting on line 716).
> We apologize for the confusion.  You are correct.
> $\tilde{F}|\_{\mathcal{L}}$ is the restriction of the domain of the $\tilde{F}$ to $\mathcal{L}$ and $\hat{G}(z)|\_{\mathcal{L}}$ is the projection of $\hat{G}$ onto $\mathcal{L}$. Before Lemma 9, we will define the notion $\tilde{F}|\_{\mathcal{L}}$ as the function restriction, define $P\_{\mathcal{L}}$ as the projection operator and replace $\hat{G}(z)|\_{\mathcal{L}}$ with $P\_{\mathcal{L}}(\hat{G}(z))$ to clarify.
> We will also include this notation in a notation table in the appendix.
>
> 4. We apologize for the confusion that caused.  We re-used the notation "$\mathcal{L}$" in those places for different affine spaces.  First, we remark that each definition is correct within its respective local scope (Algorithm 2, Lemma 9, Theorem 1). Second, we will revise the notation, fixing $\mathcal{L} = \operatorname{aff}(\mathcal{K})$ and defining $\mathcal{L}_0 = \operatorname{aff}(\mathcal{K}) - z$ (for some $z \in \mathcal{K}$) to distinguish between the two affine spaces.
> We will also include this notation in a notation table in the appendix.
>
> 5. Thank you for the suggestion.  We agree this point should be more clearly motivated.
> Please refer to the "global rebuttal" above for a detailed explanation.
>
> 6. We agree.  Including lower bounds (when known) would provide important context for the sample complexity and regret bounds (for offline and online settings respectively) we and the SOTA have obtained.
> We will mention lower bounds in the introduction and include a discussion on those results in Appendix A: Details of Related Works.  (In Appendix A.1., we pointed out the optimality of the approximation ratios for two cases (mentioned in the following), but will make clear there and the introduction those are the only cases with tight bounds.)
> For reference, we will briefly summarize lower bounds and hardness results. The approximation coefficients are optimal for the case (i) where the function is monotone and the constraint set contains the origin and the case (ii) where the function is non-monotone and the constraint set is a general convex set. For monotone functions over general convex set, we conjecture that $1/2$ is the optimal coefficient (See the "global rebuttal" above for more details).
> For non-monotone functions over downward closed convex sets, the $1/e$ coefficient is known to be sub-optimal. (See the recent paper "Continuous Non-monotone DR-submodular Maximization with Down-closed Convex Constraint" by Chen et al, arXiv.org:2307.09616, that obtains 0.385 coefficient) and the best known upper bound so far is 0.491 (See "Submodular maximization by simulated annealing" by Gharan et al. 22nd ACM-SIAM, 2011)
> With a stochastic gradient oracle, when maximizing a monotone function over a downward closed convex set, the lower bound on oracle complexity is $1/\varepsilon^2$ (see "Stochastic Continuous Greedy ++: When Upper and Lower Bounds Match" by Karbasi et al. in NeurIPS 2019)
> We expect the algorithm to be efficient in the deterministic gradient oracle case for all where the approximation coefficient is optimal. This intuition is based on the fact that $O(1/\varepsilon)$ is a fundamental barrier of linear programming based methods in general within the context of convex optimization. (See "Conditional Gradient Methods", Braun et al, arXiv:2211.14103v2, Section 2.1.2)
>
> 7. We will revise the caption for Table 1 to include $h = \min_{z \in \mathcal{K}} \|z\|_\infty$.
>
> 8. We will follow your suggestions, adding a notation table to the appendix and adding verbal reminders of what notation refers to throughout the paper (esp. in the appendices).

---

> > ### Comment · Reviewer_GwvC · 2023-08-10
> > **Response to rebuttal**
> >
> > Thank you for the detailed answer. I have some follow up questions/comments:
> > 1. Given these modifications, the algorithm requires knowing the dimension of the affine hull of the constraint set, which might not be easy to compute for complicated constraint sets. This limitation should be stated clearly.
> > 2. So this claim requires that D (which corresponds to the constraint set K) to be a manifold? This should also be stated clearly then.
> > 5. I did not see any discussion about the motivation for restricting the oracle to query only feasible points in the "global rebuttal" above.
> >
> > One other clarity issue I noticed: $\hat{F}$ is defined as a function of two variables $z, x$ in Section 3.1, but later one it is used as a function of a single variable (for example in Algorithm 1), without clarification of how it relates to the original definition.

---

> > > ### Author Response · Authors · 2023-08-11
> > >
> > > Thank you for your detailed review.
> > >
> > > 1. We agree, and will add that in the final version.
> > >
> > > 2. We note that lines 671-672 did not play any role it the proof and were meant for clarification.
> > > However, we acknowledge that they are not as clarifying as we have hoped and we will remove them.
> > > To answer your question, lines 671-672 should be revise to say "$\nabla F(z)$ lives within the linear space $A_0$ for all $z \in D$ where $\mathbb{B}\_\delta^A(z) \subseteq D$."
> > > Please note that the last line in our previous response to Q.2 should be:
> > > "As a special case, if $M \subseteq A$ where $A$ is an affine space, then for all $z \in M$ the tangent space $T_z(M) \subseteq A_0$, where $A_0 = A - x$ for some $x \in A$. Therefore we have $\nabla F(z) \subseteq A_0$ for all $z \in M$."
> > > Also note that we do not require $D$ to be a manifold.
> > > However, the set $\bigcup\_{\\{z \in D \mid \mathbb{B}\_\delta^A(z) \subseteq D\\}} \mathbb{B}\_\delta^A(z) \subseteq D$ is a union of $k$-dimensional spheres, where $k = dim(\operatorname{aff}(D))$, and so it is a $k$-dimensional manifold.
> > >
> > > 3. Sorry, due to character limits, we moved that to the response to Weakness 2 to Reviewer Dxab - though we write it again here.
> > > **Motivation for (feasible) value oracle queries:** We will revise the introduction to better motivate the importance of developing optimization methods for value oracle queries, including just over the feasible region. We first highlight two points and then discuss application motivations.
> > > (1) *Offline-to-online adaptations*
> > > For online optimization problems, when only bandit feedback is available (it is typically a strong assumption that semi-bandit or full-information feedback is available), then the agent must be able to learn from stochastic value oracle queries over the feasible actions action. By designing offline algorithms that only query feasible points, we made it possible to convert those offline algorithms into online algorithms. In fact, because of how we designed the offline algorithms, we are able to access them in a black-box fashion for online problems when only bandit feedback is available. *Note that previous works on DR-submodular maximization with bandit feedback in monotone settings (i.e. [29] and arXiv:2305.12402) explicitly assume that the convex set contains the origin.*
> > > (2) *More precise characterizations of inherent challenges underlying approximation guarantees*
> > > As noted in the "global rebuttal" above, in developing a unifying framework where we took care to characterize how powerful the oracles were, we identified the underlying causes for an approximation gap between gradient ascent and Frank-Wolfe methods.
> > > **Applications:**
> > > We will revise the paper by discussing "classic" example applications that prior works (like [arXiv:2006.13474]) have shown to be instances of constrained DR-submodular maximization, such as influence/revenue maximization, facility location, and non-convex/non-concave quadratic programming, as well as more recently identified applications like serving heterogeneous learners under networking constraints [arXiv:2201.04830] and joint optimization of routing and caching in networks [arXiv:2302.02508]. We will comment on how strong/mild assuming availability of anything more powerful than a value oracle over the feasible region is. For many problems, the ability to evaluate gradients directly requires strong assumptions about problem-specific parameters.
> > > Influence maximization and profit maximization form a family of problems that model choosing advertising resource allocations to maximize the expected number of customers, where there is an underlying diffusion model for how advertising resources spent (stochastically) activate customers over a social network. For common diffusion models, the objective function is known to be DR-submodular (see for instance [arXiv:2006.13474] or [arXiv:2212.06646]). The revenue (expected number of activated customers) is a monotone objective function; total profit (revenue from activated customers minus advertising costs) is a non-monotone objective.
> > > One significant challenge with these problems is that the objective function (and the gradients) cannot be analytically evaluated for general (non-bipartite) networks, even if all the underlying diffusion model parameters are known exactly.
> > > The mildest assumptions on knowledge/observability of the network diffusions for offline variants (respectively actions for online variants), especially fitting for user privacy and/or third-party access, leads to instantiations of queries as the agent selecting an advertising allocation within the budget (i.e., feasible point) and observing a (stochastic) count of activated customers. This corresponds to stochastic value oracle queries over the feasible region (respectively bandit feedback for online variants).
> > >
> > > 4. We will clarify in Section 3.1 that when $\hat{F}$ is taking one variable, we are treating it as a random variable.

---

> > > > ### Comment · Reviewer_GwvC · 2023-08-11
> > > >
> > > > - I thought the claim on lines 671-672 is needed for example for $O(\nabla \tilde F'(0))$ to be well defined, but I can see that in this case this can be verified from the definition of $F'$. Thanks for the detailed explanation!
> > > >
> > > > I will raise my score by one point.

---

> > > > > ### Author Response · Authors · 2023-08-12
> > > > >
> > > > > Thank you for your detailed and prompt responses and for increasing the score!

---

### Author Rebuttal · Authors · 2023-08-10

We highlight our technical contributions (in addition to contributions in obtaining new guarantees for numerous offline and online settings as well as unifying algorithm design and analysis among several prior works).

1. **Our procedure is the first Frank-Wolfe type algorithm for analyzing monotone functions over general a convex set when the oracle is only allowed to query within the feasible set, for any type of oracle for the objective function (exact/stochastic value/gradient).**
Note that the algorithm in this case is the same as the algorithm for non-monotone general convex case, only with a different step-size.
The main challenge here was recognizing that the analysis for the non-monotone general convex setting and the monotone setting where the convex set contains the origin could be combined to prove regret bounds for this algorithm.

2. **A new construction procedure of a shrunk constraint set that allows us to work with lower dimensional feasible sets when given a value oracle, resulting in the first results on general lower dimensional feasible sets given a value oracle.**
*Please refer to Q.1 in the response to Reviewer L9XA for details about differences with prior works.*

3. **Our work sheds light on a previously unexplained gap in approximation guarantees for monotone DR-submodular maximization.**
(We briefly mentioned some of the following in our paper, but will revise the main section and appendices to make the following clear.)
Specifically, some prior works (enumerated below) studying monotone DR-submodular maximization over general convex sets obtained guarantees of $1/2$ while others obtained $1-1/e$.
In [16], a $1/2$ guarantee was obtained by a projected gradient ascent method and this was shown by proving that the algorithm tends to a stationary point and proving that any stationary point is at least $1/2$ as good as the optimal point. Moreover, they construct examples with stationary points that are no better that $1/2$ of the optimal point.
The $1-1/e$ guarantee was reported for Frank-Wolfe methods, which (superficially) suggests that the gap may be due to algorithm or analysis differences. However, in carrying out our work on developing a unified framework, we identified that the gap was not attributable to algorithm or analysis differences, but instead due to queries to infeasible points by the Frank-Wolfe methods (i.e. they were solving different problems).
A key ingredient to obtain $1-1/e$ was the ability to query the (gradient) oracle within the convex hull of $\mathcal{K}\cup\{0\}$.
(Note that this is true for both Frank-Wolfe based methods and for projection based methods. Please refer to response to Question 1 of reviewer f6HN for more details about the projection based methods.)
For monotone submodular maximization over general convex sets (not necessarily containing the origin), we can only guarantee a coefficient of $1/2$, both for Frank-Wolfe type methods (our work) and projection based methods (i.e. [16]).
Moreover, it is evident from our proofs that the case of maximizing a monotone function over a general convex region  with the origin infeasible ($0 \not \in \mathcal{K}$) is the only case where the starting point ($\mathbf{z}_1$ in our paper) of the algorithm does not matter.
To the best of our knowledge, in every paper where the $1/2$ approximation coefficient and $1-1/e$ approximation coefficient in the monotone setting are compared, the comparison was unwittingly between problems that are inherently mathematically different:
[16] and [8]  in experiments and main text;
[7] and [9] in experiments; [30], [23], and [13] in related work section, [22] in the introduction  and Table 2, ”Stochastic continuous submodular Maximization: Boosting via non-oblivious function” by Zhang et al. ICML 2022 in the main claim; and ”Fast First-Order Methods for Monotone Strongly DR-Submodular Maximization.” Fazel et al. (ACDA23), 2023  in the main claims.
In other words, the $1/2$ approximation could very well be optimal in its own setting. We will add this explanation and the following conjecture to the final version.
**Conjecture**: *The problem of maximizing a monotone DR-submodular continuous function subject to a general convex constraint, where the oracle access is limited to the feasible region, is NP-hard. For any $\epsilon > 0$, it cannot be approximated in polynomial time within a ratio of $1/2 + \epsilon$ (up to low-order terms), unless $RP = NP$.*

**Minor correction to Tables**
In Table 2, we cite 4 papers, but 2 of them ([29] and [30]) are referring to a method known as Mono-Frank-Wolfe, which is not for online stochastic setting since they choose one point but query feedback for another point.   The feedback model Mono-Frank-Wolfe relies on is more informative than the semi-bandit gradient $\nabla F$ feedback included in Table 2. Moreover, the paper [8] mentioned in Table 2 should be replaced with [16] since in the special case considered in Table 2, both algorithms will be the same and it was first proposed by [16].
As we have discussed in rebuttals below, the papers mentioned by reviewers L9XA and f6HN should also be added.

---

### Decision · Program_Chairs · 2023-09-21

**Decision:**

Accept (poster)

**Comment:**

The paper addresses a significant problem by studying the maximization of stochastic DR-submodular functions across a diverse range of settings. These settings encompass factors such as monotonicity, feasible region characteristics, oracle access, and the nature of the oracle itself. The authors propose a unified approach grounded in the Frank-Wolfe meta-algorithm. Their contribution includes establishing oracle complexity guarantees, reducing computational complexity through innovative techniques, and matching guarantees in specific settings. Additionally, the paper extends its focus to online versions of the problem, showcasing algorithms with enhanced regret bounds under bandit and semi-bandit feedback scenarios.

I find the paper to be a commendable work, offering a strong and well-founded theoretical contribution that builds upon existing research in this domain. The authors' approach is notably comprehensive, addressing multiple dimensions of the problem and presenting novel insights.

Considering the above, I would recommend accepting the paper for inclusion in the conference proceedings.